# LEARNING NEURAL IMPLICIT FUNCTIONS AS OBJECT REPRESENTATIONS FOR ROBOTIC MANIPULATION

## ABSTRACT

Robotic manipulation planning is the problem of finding a sequence of robot configurations that involves interactions with objects in the scene, e.g., grasp, placement, tool-use, etc. To achieve such interactions, traditional approaches require hand-designed features and object representations, and it still remains an open question how to describe such interactions with arbitrary objects in a flexible and efficient way. Inspired by recent advances in 3D modeling, e.g. NeRF, we propose a method to represent objects as neural implicit functions upon which we can define and jointly train interaction constraint functions. The proposed pixel-aligned representation is directly inferred from camera images with known camera geometry, naturally acting as a perception component in the whole manipulation pipeline, while at the same time enabling sequential robot manipulation planning.

## 1 INTRODUCTION

Intelligent agents should be able to interact with objects in the environment, such as grasping and placing an object, or more general tool-use, to achieve a certain goal. In robotics, such instances are formalized as manipulation planning, a type of a motion planning problem that solves not only for the robot's own movement but also for the objects' motions subject to interaction constraints. Traditional approaches represent objects using meshes or combinations of shape primitives and describe interactions as hand-crafted constraints in terms of that representation. The approach of using such traditional geometric representations has long-standing limitations in terms of their perception and generalizing to large varieties of objects and interaction modes: (i) The representations have to be inferred from raw sensory inputs like images or point clouds – raising the fundamental problem of perception and shape estimation. However, if the aim is manipulation skills, the hard problem of precise shape estimation might be unnecessary to predict accurate interaction features[1], and an end-to-end object representation might be more appropriate than a standard perception pipeline. (ii) With increasing generality of object shapes and interaction, the complexity of representations grows and hand-engineering of the interaction features becomes inefficient.

*What is a good representation of an object*? Considering the representation will be used to predict interaction features, we expect it to encode primarily task-specific information rather than only geometric. And some of the information should to be shared across different interaction modes. In other words, good representations should be task-specific so that the feature prediction can be simplified and, at the same time, be task-agnostic to enable synergies between the features. E.g., mug handles are called handles because we can handle the mug through them and also, once we learn the notion of a handle, we can interact with the mug through them in many different ways. From the perception aspect, good representations should be easy to infer from raw sensory inputs and should be able to trade their accuracy off in favor of the feature prediction.

To this end, we propose a novel data-driven approach to learning interaction features. The proposed feature prediction scheme is illustrated in Fig. 1. The whole pipeline is trained end-to-end directly with the task supervisions so as to make the representation and perception *task-specific* and thus to simplify the interaction prediction. The object representation acts as a bottleneck and is shared across multiple feature predictions so that the *task-agnostic* representations can emerge. Particularly, the object representation is a neural implicit function over the 3D space (Park et al., 2019; Mildenhall et al., 2020) upon which equality constraint features are trained. The proposed neural implicit

---

[1]We call an interaction constraint function an interaction feature; when used as equality constraints, the interaction features, analogous to energy potentials, return zero when feasible and non-zero otherwise.

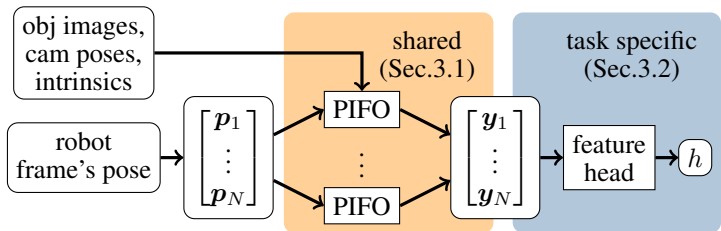

Figure 1: The interaction feature prediction scheme

function is pixel-aligned: The function takes images from multiple cameras as input (e.g. stereo) and, assuming known camera poses and intrinsics, the latent representation at a certain spatial location is directly related to pixels of the images. Once learned, the interaction features can be used by a typical constrained optimal control framework to plan dexterous object-robot interaction. We adopt Logic-Geometric Programming (LGP) (Toussaint et al., 2018) as an optimization-based manipulation planning framework and show that this learned-feature based planning enables to compute trajectories that involve various types of interaction modes only from images. Due to the representations' generalization, the learned features are directly applicable to manipulation tasks involving unseen objects. To summarize, our main contributions are

- To represent objects as neural implicit functions upon which interaction features are trained,
- An image-based manipulation planning framework with the learned features as constraints,
- Comparison to non pixel-aligned, non implicit function, and geometric representations,
- Demonstration in various manipulation scenarios ranging from simple pick-and-hang [videos] to longer-horizon manipulations [videos] and zero-shot imitations [videos].

## 2 RELATED WORK

### 2.1 NEURAL IMPLICIT REPRESENTATIONS IN 3D MODELING AND VIEW SYNTHESIS

Neural implicit representations have recently gained increasing attention in 3D modeling. The core idea is to encode an object or a scene in the weights of a neural network, where the network acts as a direct mapping from 3D spatial location to an implicit representation of the model, such as occupancy measures (Mescheder et al., 2019; Songyou Peng, 2020) or signed distance fields (Park et al., 2019; Gropp et al., 2020; Atzmon & Lipman, 2020). In contrast to explicit representations like voxels, meshes or point clouds, the implicit representations don't require discretization of the 3D space nor fixed shape topology but rather continuously represent the 3D geometry, thereby allowing for capturing complex shape geometry at high resolutions in a memory efficient way.

There have been attempts to associate these 3D representations with 2D images using the principle of camera geometry. Exploiting the camera geometry in a forward direction, i.e., 2D projection of 3D representations, yields a differentiable image rendering procedure and this idea can be used to get rid of 3D supervisions. For example, Sitzmann et al. (2019); Niemeyer et al. (2020); Yariv et al. (2020); Mildenhall et al. (2020); Henzler et al. (2021); Reizenstein et al. (2021) showed that the representation networks can be trained without the 3D supervision by defining a loss function to be difference between the rendered images and the ground-truth. Another notable application of this idea is view synthesis. Based on the differentiable rendering, Park et al. (2020); Chen et al. (2020); Yen-Chen et al. (2021) addressed unseen object pose estimation problems, where the goal is to find object's pose relative to the camera that produces a rendered image closest to the ground truth. By conditioning 3D representations on 2D input images, one can expect the amortized encoder network to directly generalize to novel 3D geometries without requiring any test-time optimization. This can be done by introducing a bottleneck of a finite-dimensional *global* latent vector between the images and representations, but these global features often fail to capture fine-grained details of the 3D models (Songyou Peng, 2020). To address this, the camera geometry can be exploited in the inverse direction to obtain pixel-aligned *local* representations, i.e., 3D reprojection of 2D image features. Saito et al. (2019) and Xu et al. (2019) showed that the pixel-aligned methods can establish rich latent features because they can easily preserve high-frequency components in the input images. Also, Yu et al. (2021) and Trevithick & Yang (2021) incorporated this idea within the view-synthesis framework and showed that their convolutional encoders have strong generalizations.

While the above work investigates neural implicit functions to model shapes or appearances, we train them to model physical interaction feasibility and thereby to provide a differentiable constraint model for robot manipulation planning.

## 2.2 Object/Scene Representations for Robotic Manipulations

Several works have proposed data-driven approaches to learning object representations and/or interaction features which are conditioned on raw sensory inputs, especially for grasping of diverse objects. One popular approach is to train discriminative models for grasp assessments. For example, ten Pas et al. (2017); Mahler et al. (2017); Van der Merwe et al. (2020) trained a neural network that, for given candidate grasp poses, predicts their grasp qualities from point clouds. In addition, Breyer et al. (2020); Jiang et al. (2021) proposed 3D convolutional networks that take as inputs a truncated SDF and candidate grasp poses and return the grasp affordances. Similarly, Zeng et al. (2020b;a) addressed more general manipulation scenarios such as throwing or pick-and-place, where a convolutional network outputs a task score image. On the other hand, neural networks also have been used as generative models. For example, Mousavian et al. (2019) and Murali et al. (2020) adopted the approach of conditional variational autoencoders to model the feasible grasp pose distribution conditioned on the point cloud. Sundermeyer et al. (2021) proposed a somewhat hybrid method, where the network densely generates grasp candidates by assigning grasp scores and orientations to the point cloud. You et al. (2021) addressed the object hanging tasks from point clouds where the framework first makes dense predictions of the candidate poses among which one is picked and refined. Compared to these works, our framework takes advantage of a trajectory optimization to jointly optimize an interaction pose sequence instead of relying on exhaustive search or heuristic sampling schemes, thus not suffering from the high dimensionality nor the combinatorial complexity of long-horizon planning problems.

Another important line of research is learning and utilizing keypoint object representations. Manuelli et al. (2019); Gao & Tedrake (2021); Qin et al. (2020); Turpin et al. (2021) represented objects using a set of 3D semantic keypoints and formulated manipulation problems in terms of such the keypoints. Similarly, Manuelli et al. (2020) learned the object dynamics as a function of keypoints upon which a model predictive controller is implemented. Despite their strong generalizations to unseen objects, the keypoint representations require semantics of the keypoints to be predefined. The representation part of our framework is closely related to dense object descriptions proposed by Florence et al. (2018; 2019). The idea is to train fully-convolutional neural networks that maps a raw input image to pixelwise object representations which directly generalize to unseen objects. Our proposed framework can be seen as an extension of this pixelwise representation to dense representations over the 3D space which is learned by the task supervisions and can be seamlessly integrated into general sequential manipulation planning problems. Another recent related work was proposed by Yuan et al. (2021), where the learned object-centric representations are used to predict the symbolic predicates of the scene which in turn enables symbolic-level task planning. In contrast, our framework predicts the task feasibility given a robot configuration and enables trajectory optimization of the lower-level continuous motions.

## 3 Interaction Feature Prediction via Implicit Representation

Given $N_{\text{view}}$ images with their camera poses/intrinsics, $\{(\mathcal{I}^1, \boldsymbol{T}^1, \boldsymbol{K}^1), ..., (\mathcal{I}^{N_{\text{view}}}, \boldsymbol{T}^{N_{\text{view}}}, \boldsymbol{K}^{N_{\text{view}}})\}$, we define an interaction feature as a neural implicit function:

$$h = \phi_{\text{task}}(\boldsymbol{q}; \{(\mathcal{I}^1, \boldsymbol{T}^1, \boldsymbol{K}^1), ..., (\mathcal{I}^{N_{\text{view}}}, \boldsymbol{T}^{N_{\text{view}}}, \boldsymbol{K}^{N_{\text{view}}})\}), \quad (1)$$

where $\boldsymbol{q} \in SE(3)$ is the pose of the robot frame interacting with the object. As shown in Fig. 1, the feature prediction framework consists of two parts: the representation network, which we call a backbone, and the feature head networks. The backbone serves as an implicit functional representation of an object, which, conditioned on a set of posed images, outputs $d$-dimensional representation vectors at queried 3D spatial locations. The interaction feature predictions are made through the feature heads, where each head is fed on a set of representation vectors obtained by querying the backbone at a set of key interaction points. While the multiple feature heads separately model different interactions, the backbone is shared across the tasks, making it learn more general object representations. The rest of this section will be devoted to introduce each module in detail.

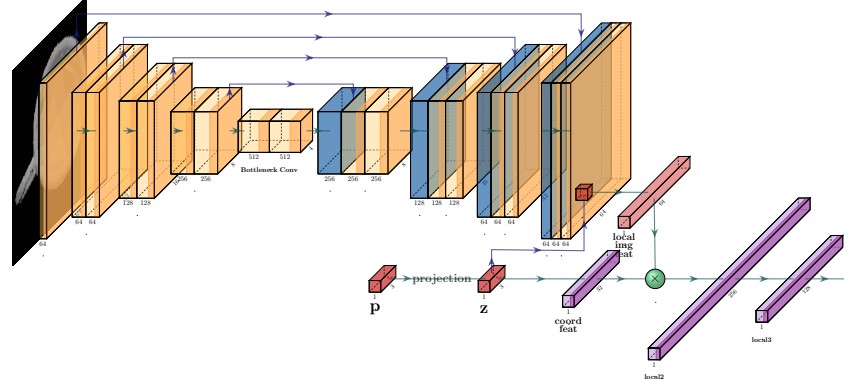

Figure 2: Image encoder and 3D reprojector of the backbone network.

## 3.1 PIXEL-ALIGNED IMPLICIT FUNCTIONAL OBJECT REPRESENTATION (PIFO)

Given $N_{\text{view}}$ posed images, we define a functional representation of an object as a mapping:

$$\psi(\boldsymbol{p}; \{(\mathcal{I}^1, \boldsymbol{T}^1, \boldsymbol{K}^1), ..., (\mathcal{I}^{N_{\text{view}}}, \boldsymbol{T}^{N_{\text{view}}}, \boldsymbol{K}^{N_{\text{view}}})\}) = \boldsymbol{y}, \tag{2}$$

where $\boldsymbol{p} \in \mathbb{R}^3$ and $\boldsymbol{y} \in \mathbb{R}^d$ are a queried 3D position and a representation vector at that point, respectively. The representation network consists of three parts: image encoder, 3D reprojector, and feature aggregator. The first two modules compute a representation vector for each image, as depicted in Fig. 2, and the last one combines them into one vector.

**Image Encoder:** This module takes as input an image and computes a feature image. In order to capture both local and global information in the image, we adopted the U-net architecture (Ronneberger et al., 2015), especially with ResNet-34 (He et al., 2016) as its downward path and two residual $3 \times 3$ convolutions followed by up-convolution as the upward path, i.e.,

$$\mathcal{F}^n = UNet(\mathcal{I}^n), \ \forall n \in \{1, ..., N_{\text{view}}\}. \tag{3}$$

**3D Reprojector:** To endow the network with the multi-view consistency, all the 3D operations are performed in the view space. The 3D reprojector first converts a queried point, $\boldsymbol{p}$, to the image coordinate including depth, $\pi(\boldsymbol{p}; \boldsymbol{T}, \boldsymbol{K}) = \boldsymbol{z} \in \mathbb{R}^3$ and then uses the projected point to extract the local image feature from the feature image, $\mathcal{F}$, via bilinear interpolation. Finally, the extracted feature and the coordinate feature, which is necessary to handle out-of-image query points, are passed to a couple of fully connected layers to get a representation vector for a single image, i.e.,

$$\boldsymbol{y}^n = MLP(\mathcal{F}^n(\boldsymbol{z}^n), \boldsymbol{z}^n), \ \boldsymbol{z}^n = \pi(\boldsymbol{p}; \boldsymbol{T}^n, \boldsymbol{K}^n), \ \forall n \in \{1, ..., N_{\text{view}}\}. \tag{4}$$

**Feature Aggregator:** The feature aggregation from the multiple views should be permutation-invariant. While there are many options, like summation, averaging, or more sophisticated attention mechanisms, we simply take the averaging operation for it, i.e. $\boldsymbol{y} = \frac{1}{N_{\text{view}}} \sum_n \boldsymbol{y}^n$.

## 3.2 FEATURE PREDICTION

A feature head network predicts the interaction constraint value of a robot frame's pose via the object representation. We first attach a set of keypoints to the robot frame and query the backbone at those keypoint positions, i.e., $\forall i \in \{1, ..., N_{\text{keypoint}}\}$,

$$\boldsymbol{y}_i = \psi(\boldsymbol{p}_i; \{(\mathcal{I}^n, \boldsymbol{T}^n, \boldsymbol{K}^n)|n \in \{1, ..., N_{\text{view}}\}\}), \ \boldsymbol{p}_i = \boldsymbol{R}(\boldsymbol{q})\hat{\boldsymbol{p}}_i + \boldsymbol{t}(\boldsymbol{q}), \tag{5}$$

where $\hat{\boldsymbol{p}}_i$ is $i^{\text{th}}$ keypoint's local coordinate, and $\boldsymbol{R}(\boldsymbol{q})$ and $\boldsymbol{t}(\boldsymbol{q})$ denote the rotation matrix and the translation vector, respectively. The feature head then takes as input the resulting representation vectors and predicts a feature value through a couple of fully connected layers, i.e.,

$$h = MLP(\boldsymbol{y}_1, ..., \boldsymbol{y}_{N_{\text{keypoint}}}). \tag{6}$$

Note that when the considered feature is an SDF, only a single point needs to be queried, i.e., $N_{\text{keypoint}} = 1$, so the whole architecture reduces to the surface reconstruction in Saito et al. (2019).

# 4 TRAINING

In this paper, we consider manipulation scenarios where a robot arm, Franka Emika Panda, or two have to pick and hang mugs on hooks. The environment contains mugs having different shapes and hooks on which a mug can be hung. To formulate this manipulation problem, three interaction features are considered: an SDF feature for collision avoidance and grasping/hanging features.

## 4.1 DATA GENERATION

We took 131 mesh models of mugs from ShapeNet (Chang et al., 2015) and convex-decomposed those meshes. The meshes are translated and randomly scaled so that they can fit in a bounding sphere with a radius of $10 \sim 15$ cm at the origin. For each mug, we created datasets of the posed images and each interaction as follows.

**Posed Images:** The posed image data consists of 100 images ($128 \times 128$) with the corresponding camera poses and intrinsic matrices generated by an OpenGL renderer. Azimuths and elevations of the cameras are sampled such that they can uniformly be distributed on the surface of a sphere, while their distances from the object center are randomly chosen. The azimuth, elevation and distance fully determine the camera's positions and making them upright and face the object center gives the camera's orientations. For the intrinsics, we used $fov = 2 \arcsin(d/r)$, where $d$ is the camera distance from the object center and $r$ is the radius of the object's bounding sphere, so that the entire object appears in the image. Lighting is also randomized.

**SDF:** We sampled 12,500 3D positions and computed their signed distance values using the mesh-to-sdf library (Kleineberg, 2021). Following the approach of DeepSDF (Park et al., 2019), we sampled more aggressively near the object surface to foster the learning of the object geometry.

**Grasping & Hanging:** We obtained 1,000 feasible grasping and hanging poses of the gripper and the hook, respectively. For grasping, we used an antipodal sampling scheme, similarly to Eppner et al. (2021), to create candidate gripper poses and checked their feasibility using Bullet (Coumans & Bai, 2016–2021). For hanging, we randomly sampled collision-free hook poses and checked if it's kinematically trapped by the mug in the directions perpendicular to the hook's main axis.

Fig. 8 shows some rough looks of the generated data. In the end, we have a dataset of:

$$\mathcal{D} = \left\{ \left( \mathcal{I}^{1:100}, \boldsymbol{T}^{1:100}, \boldsymbol{K}^{1:100}, \boldsymbol{p}^{1:12500}, SDF^{1:12500}, \boldsymbol{q}_{\text{grasp}}^{1:1000}, \boldsymbol{q}_{\text{hang}}^{1:1000} \right)^{(i)} \right\}_{i=1}^{131}, \quad (7)$$

which we divided into 78 training, 25 validation and 28 test sets.

## 4.2 DATA AUGMENTATION AND LOSS FUNCTION

**Data Augmentation:** While randomizing the azimuth, elevation and distance of the camera provides all possible appearances of the object, it still cannot account for the roll angles of the camera and off-centered images. To show the network all possible images that it can encounter when deployed later and to mitigate the size-ambiguity issue, we propose to use a data augmentation technique based on Homography warping: In each iteration, for a randomly sampled set of images, we artificially perturb the roll angle of each camera and the estimated object center position (at which the cameras are looking). Also, $fov$ is modified as if the radius of the bounding sphere is 15 cm so that smaller objects can appear smaller in the transformed images. This results in new rotation matrices, $\hat{\boldsymbol{R}}$, and intrinsic matrices, $\hat{\boldsymbol{K}}$, of the cameras. We then compute the corresponding Homography transformation and warp the images accordingly (details in Appendix A.1):

$$\mathcal{W}(\hat{\boldsymbol{R}}, \hat{\boldsymbol{K}}) : \begin{bmatrix} u \\ v \\ 1 \end{bmatrix} \mapsto w \hat{\boldsymbol{K}} \hat{\boldsymbol{R}}^T \boldsymbol{R} \boldsymbol{K}^{-1} \begin{bmatrix} u \\ v \\ 1 \end{bmatrix}. \quad (8)$$

Random cutouts are also applied to address the object occlusion. Fig. 9 depicts how this image augmentation works. For grasping and hanging, we generate 6D random poses $\hat{\boldsymbol{q}}_{\text{task}} \sim \mathcal{P}$ in each iteration[2] and, similarly to Atzmon & Lipman (2020), set the training target to be unsinged distances in SE(3) to the set of the feasible poses: $d(\boldsymbol{q}; \boldsymbol{q}_{\text{task}}^{1:N_{\text{task}}}) = \min_{i=1,\dots N_{\text{task}}} ||\boldsymbol{q} - \boldsymbol{q}_{\text{task}}^i||_2$.

---

[2]To encourage more precise prediction around the constraint manifolds, we used a weighted sum of a feasible pose and a random pose $\hat{\boldsymbol{q}}_{\text{task}} = t\boldsymbol{q}_{\text{feasible}} + (1-t)\boldsymbol{q}_{\text{rand}}$, $t \sim \mathcal{U}(0,1)$ where the position of $\boldsymbol{q}_{\text{rand}}$ is from the normal distribution and its quaternion is sampled uniformly.

**Training:** For the overall network training, we first choose a minibatch of mugs from which a subset of augmented images with their camera poses and intrinsics, $\left(\hat{\mathcal{I}}^{1:N_{\text{views}}}, \hat{\boldsymbol{T}}^{1:N_{\text{views}}}, \hat{\boldsymbol{K}}^{1:N_{\text{views}}}\right)$, a subset of SDF data, $\left(\boldsymbol{p}^{1:N_{\text{SDF}}}, SDF^{1:N_{\text{SDF}}}\right)$, and the grasping/hanging data, $\left(\hat{\boldsymbol{q}}_{\text{task}}^{1:N_{\text{task}}}, d_{\text{task}}^{1:N_{\text{task}}}\right)$, are sampled. The images are encoded only once per iteration and then the SDF, grasping, hanging features are queried at the sampled points and poses. The overall loss is given as $\mathcal{L}_{\text{total}} = \mathcal{L}_{\text{sdf}} + \mathcal{L}_{\text{grasp}} + \mathcal{L}_{\text{hang}}$, where a typical $L1$ loss is used for SDFs, i.e. $\mathcal{L}_{\text{sdf}} = \frac{1}{N_{\text{SDF}}} \sum_{i=1}^{N_{\text{SDF}}} |\phi_{\text{sdf}}(\boldsymbol{p}^i) - SDF^i|$, and the sign-agnostic $L1$ loss in (Atzmon & Lipman, 2020) for grasping and hanging, i.e., $\mathcal{L}_{\text{task}} = \frac{1}{N_{\text{task}}} \sum_{i=1}^{N_{\text{task}}} \left| |\phi_{\text{task}}(\hat{\boldsymbol{q}}_{\text{task}}^i)| - d_{\text{task}}^i \right|$, $\forall \text{task} \in \{\text{grasp}, \text{hang}\}$.[3] The feature head and backbone are trained end-to-end. Specifically, we used $N_{\text{views}} = 4$, $N_{\text{SDF}} = 300$, $N_{\text{grasp}} = 100$, $N_{\text{hang}} = 100$. As shown in Fig. 11, the grasp and hang interaction points are defined as $(3 \times 3 \times 3)$ grid points around the gripper center and as 5 points along the hook's main axis, respectively.

# 5 SEQUENTIAL MANIPULATION PLANNING WITH LEARNED FEATURES

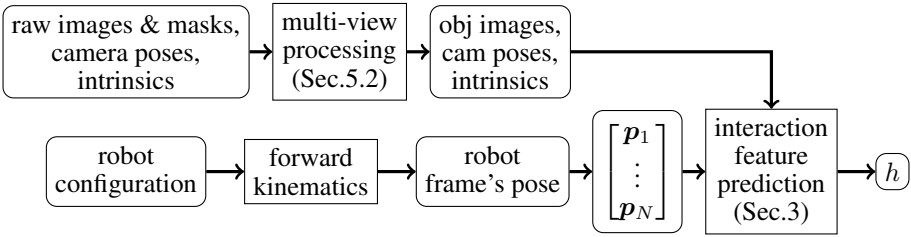

Figure 3: The proposed interaction feature prediction scheme for manipulation planning

In order to compute a full trajectory of the robot and the objects that it interacts with, the learned features can be integrated into any constraint-based trajectory optimization and manipulation planning framework, using the features as differentiable interaction constraints. In this work, we adopt Logic-Geometric Programming (LGP) (Toussaint et al., 2018) as an optimization-based manipulation planning framework. In typical manipulation scenes, cameras are equipped such that their views cover a wide range of the environment. As shown in Fig. 3, we warp raw images of the entire scene to get object-centric images and corresponding camera extrinsics/intrinsics and compute the robot frame's pose via forward kinematics to feed them into the network. Section 5.1 presents how the learned constraint functions are integrated into sequential manipulation planning problems and Section 5.2 discusses the proposed warping procedure.

## 5.1 LOGIC-GEOMETRIC PROGRAMMING FOR MANIPULATION PLANNING

The core concept of manipulation planning is the rigid transformations of objects. For an object transformed by $\delta\boldsymbol{q} \in SE(3)$, we define a rigid transformation of the interaction feature as:

$$T(\delta\boldsymbol{q})[\phi_{\text{task}}](\cdot) := \phi_{\text{task}}\left(\delta\boldsymbol{q}^{-1}\cdot\right), \tag{9}$$

which is equivalent to transforming the representation as $T(\delta\boldsymbol{q})[\psi](\cdot) = \psi\left(\boldsymbol{R}(\delta\boldsymbol{q})^T\left(\cdot - \boldsymbol{t}(\delta\boldsymbol{q})\right)\right)$. By composing the forward kinematics with the feature as

$$H_{\text{task}}(\boldsymbol{x}, \delta\boldsymbol{q}) := \left(T(\delta\boldsymbol{q})[\phi_{\text{task}}] \circ FK\right)(\boldsymbol{x}), \tag{10}$$

we obtain an interaction feature as a function of a robot joint configuration $\boldsymbol{x}$ and object's rigid transformation that can be equipped into manipulation planning.

Now we are ready to formally formulate manipulation planning problems. For an $n$-joint robot and $m$ rigid objects, LGP is a hybrid optimization problem over the number of phases $K \in \mathbb{N}$, a sequence of discrete actions $\text{a}_{1:K}$ and sequences of the robot joint configurations $\boldsymbol{x}_{1:KT}$, $\boldsymbol{x} \in \mathbb{R}^n$ and the object's rigid transformations $\delta\boldsymbol{q}_{1:KT}$, $\delta\boldsymbol{q} \in SE(3)^m$. The trajectory is discretized into $T$ steps per phase. A discrete action $\text{a}_k$ describes which interaction should be fulfilled at the end of the phase $k$, i.e., which mug to pick or on which hook to hang a grasped mug, and uniquely

---

[3]For simplicity's sake, we abuse the notation of Equation 1 as $\phi_{\text{task}}(\boldsymbol{q})$ when it doesn't lead to confusion.

determines an interaction mode $s_k = \text{succ}(s_{k-1}, a_k)$, i.e., whether each mug is grasped or hung on a particular hook. Suppose that a discrete action sequence $a_{1:K}$ and the corresponding modes $s_{1:K}$ with $s_K \in \mathcal{S}_{\text{goal}}$ are proposed by a logic tree search. We define the geometric path problem as a $2^{\text{nd}}$ order Markov optimization (Toussaint, 2017):

$$\min_{\substack{\boldsymbol{x}_{1:KT} \\ \delta \boldsymbol{q}_{1:KT}}} \sum_{t=1}^{KT} f\left(\boldsymbol{x}_{t-2:t}\right), \ \text{s.t.} \forall_{k \in \{1,...,K\}} \forall_{H \in \mathbb{H}(s_k, a_k)} : H\left((\boldsymbol{x}_{t-2:t}, \delta \boldsymbol{q}_{t-2:t}^i)_{(t,i) \in \mathcal{I}_H(s_k, a_k)}\right) = 0, \quad (11)$$

where the initial joint states $\boldsymbol{x}_{-1:0}$ and objects' transformations $\delta \boldsymbol{q}_{-1:0} = 0^4$ are given. $f$ is a path cost that penalizes squared accelerations of the robot joints. $\mathbb{H}(s_k, a_k)$ is a set of path constraints the discrete state and action impose on the geometric level at each phase $k(t) = \lfloor t/T \rfloor$; these constraints include physical consistency, collision avoidance, and the learned interaction constraints that ensure the success of the discrete action $a_k$. Lastly, $\mathcal{I}_H(s_k, a_k)$ decides the time slice and object index that are subject to the constraint $H$. Appendix A.2 introduces the set of imposed constraints in detail. As all the cost and constraint terms are differentiable and their Jacobians/Hessians are sparse, we can solve this optimization problem efficiently using the Gauss-Newton optimization method.

## 5.2 MULTI-VIEW PREPROCESSING

Let $\mathcal{M}_n \in \{0,1\}^{W \times H}$ be the object masks available along with the raw images $\mathcal{I}_n$, $\forall n = 1, ..., N_{\text{cam}}$. We first solve the following optimization to find a position and radius of the minimal bounding sphere such that the warped images contain all the object pixels in the original images:

$$\min_{\boldsymbol{p} \in \mathbb{R}^3, r \in \mathbb{R}^+} r, \quad \text{s.t.} \ \forall_{(u,v) \in \{(u',v'); \mathcal{M}_n(u',v')=1, \forall n \in \{1,...,N_{\text{cam}}\}\}} : ||\mathcal{W}(\hat{\boldsymbol{R}}, \hat{\boldsymbol{K}})(u,v)||_2 < 1, \quad (12)$$

where $\hat{\boldsymbol{R}}$ can be obtained from the sphere center $\boldsymbol{p}$ and the camera position $\boldsymbol{t}$, and $\hat{\boldsymbol{K}}$ is computed as $fov = 2\arcsin(||\boldsymbol{t} - \boldsymbol{p}||_2/r)$ from which the warping $\mathcal{W}$ is defined as in A.1 or Equation 8. After solving the optimization above, we fix the camera orientations $\hat{\boldsymbol{R}}$, change the intrinsics as if the bounding sphere has a radius of 15 cm and finally warp the raw images accordingly. Fig. 12 shows the raw images from an example environment and the images warped by the multi-view processing.

# 6 EXPERIMENTS

## 6.1 PERFORMANCE OF LEARNED FEATURES

**Baselines:** The key techniques of the proposed framework are threefold: the pixel-aligned local image features, the implicit function over the 3D space as representations and the task guided learning scheme. To examine the benefits from each component, three baselines are considered. *(i) Global image features*: The first baseline still represent an object as an implicit function but the image encoder outputs a global image feature as shown in Fig. 13(b) rather than having the pixel-aligned local feature extraction; we used the ResNet-34 architecture as the image encoder and fixed the other model specifications. *(ii) Vector object representations*: The second baseline represents an object as a finite-dimensional vector instead of an implicit function; as shown in Fig. 13(c), the representation network first computes the image features from the images using ResNet-34 and the camera features from the camera parameters using a couple of fully connected layers. Two features are then passed to another couple of fully connected layers to produce the object representation vector. The feature heads take as input the frame's pose as well as the object representation vector. *(iii) SDF representations*: The last baseline uses SDFs as object representations; the network architecture for the SDF feature remains the same, but the grasping and hanging heads take as inputs a set of the keypoints' SDF values instead of the $d$-dimensional representation vectors. The SDF values are detached when passed to the grasp/hang heads so the backbone is only trained by the geometry (SDF) data.

**Evaluation Metric:** Regarding the shape reconstruction, we report the Volumetric IoU and the Chamfer distance. To measure these metrics, we first randomly sampled 4 images from the dataset and reconstructed the meshes from the learned SDF feature using the marching cube algorithm (See Fig. 10). The volumetric IoU is the ratio between the intersection and the union of the reconstructed and ground-truth meshes which is (approximately) computed on the $100^3$ grid points around the objects. To compute the Chamfer distance, we sampled 10,000 surface points from each mesh and

---

[4]Note that $\delta \boldsymbol{q}$ denotes rigid transformations applied to object's implicit representation, not absolute poses.

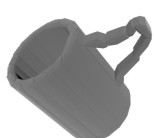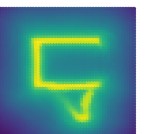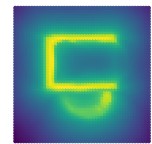  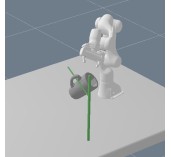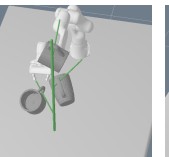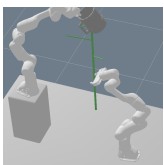

Figure 4: SDFs predicted by PIFO and the global image feature model.

Figure 5: Sequential manipulation scenarios: Single-, three-mug hanging and handover.

|  | IoU | Chamfer-$L_1$ ($\times 10^{-3}$) | Grasp+c (%) | Hang+c (%) |
|---|---|---|---|---|
| PIFO | 0.816 / 0.656 | 5.26 / 6.90 | **88.1 / 82.5** | **94.0 / 78.9** |
| Global Img. Feat. | 0.697 / 0.581 | 7.42 / 9.49 | 82.7 / 75.7 | 91.2 / 78.2 |
| Vector Obj. Repr. | 0.036 / 0.014 | 38.6 / 39.7 | 0.5 / 0.4 | 0.0 / 0.0 |
| SDF Obj. Repr. | **0.845 / 0.667** | **4.90 / 6.83** | 67.9 / 64.3 | 3.7 / 4.3 |
| PIFO (2 views) | 0.760 / 0.577 | 6.14 / 8.84 | 82.9 / 77.1 | 88.2 / 72.1 |
| PIFO (8 views) | 0.851 / 0.683 | 4.78 / 6.34 | 88.7 / 85.0 | 96.5 / 82.5 |

Table 1: Individual Feature Evaluation with 4 views (Training / Test).

averaged the forward and backward closest pair distances. To evaluate the learned task features, we solved the unconstrained optimization $\hat{q}^* = \arg\min_q ||\phi_{\text{task}}(q)||^2$, task $\in \{\text{grasp, hang}\}$ using the Gauss-Newton method. Starting from this solution, we then solved the second optimization problem by including the collision feature (details in Appendix A.3), $q^* = \arg\min_q ||\phi_{\text{task}}(q)||^2 + w_{\text{coll}}||\phi_{\text{coll}}(q)||^2$. Because the local optimization method can be stuck at local optima, we ran the algorithm from 10 random initial guesses in parallel and picked the best one. The optimized pose is then tested in simulation and the success rates are reported in Table 1.

**Result:** Table 1 shows that the SDF representation has the best shape reconstruction performance; PIFO is slightly worse, followed by the other two frameworks. On the other hand, the task performances of PIFO are significantly better than the others. The SDF representation is especially worse in the hanging task, which implies that SDFs along the line are not sufficient for the feature prediction and our task-guided representation simplifies the feature prediction. Fig. 4 depicts SDF values of an unseen mug with a complex shape handle predicted by PIFO and the global image feature model; one can observe that the global image feature model reconstructed the handle shape as being more "typical" and the pixel-aligned representation was better able to capture fine-grained details. PIFO was also tested with the different numbers of input images and it can be seen from the last two rows of the table that the more images we put in, the better performance the network shows. Tables 2 and 3 report all combinations of the metrics and the number of views.

## 6.2 SEQUENTIAL MANIPULATION PLANNING VIA LGP

As shown in Fig. 5, we considered three manipulation scenarios: Basic pick & hang, long-horizon three-mug hanging, and handover. In the first scenario, the environment contains one robot arm, one hook, one mug and 4 cameras (as in Fig. 12(a)), and the interaction modes are constrained by the discrete action sequence of [(GRASP, gripper, mug), (HANG, hook, mug)]. 10 mugs were picked from each of the training and test data sets and their initial poses are randomized.[5] When executed the optimized trajectory in the Bullet simulation, the success rates on the train and test mugs were 50 % and 40 %, respectively. If we allow the method to re-plan and execute when it failed, the success rates increased to 90% and 70%, respectively [playlist1]. The three-mug scenario consists of 6 discrete phases with [(GRASP, gripper, mug1), (HANG, M_hook, mug1), (GRASP, gripper, mug2), (HANG, U_hook, mug2), (GRASP, gripper, mug3), (HANG, L_hook, mug3)]. The handover scenario has two arms at different heights and the target hook is placed very high, requiring two arms to coordinate a handover motion; the corresponding discrete actions are [(GRASP, R_gripper, mug), (GRASP, L_gripper, mug), (HANG, U_hook, mug)]. Fig. 5 shows the last configurations of the optimized plans; we refer the readers to Figs. 16–17 and the videos [playlist2] for clearer views.

**Inverse Kinematics with Generative Models:** One important attribute of our framework is that, while most existing works train generative models that directly produce the interaction poses, ours models interactions as equality constraints which can jointly be optimized with other planning features. To see the benefits of such joint optimization, we considered the following inverse kinematics problem with a generative model: For the basic pick & hang and handover scenarios, we optimized

---

[5]Before solving the full trajectory optimization, we first optimized each feature as in Sec. 6.1 and added small regularization terms using the optimized poses to guide the optimizer away from local optima.

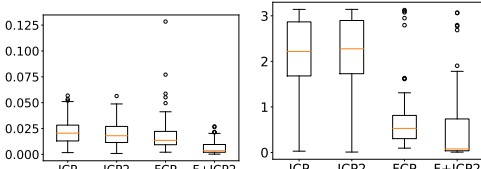
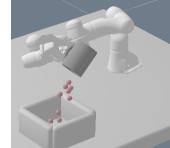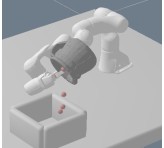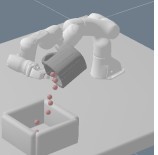

Figure 6: Position and Orientation Errors    Figure 7: (a) Reference (b-c) Imitation

each interaction pose separately as in Sec. 6.1 and checked if these individually optimized poses are kinematically feasible, i.e. whether or not the inverse kinematics problems have a solution. Even though the mug's initial pose was given such that the first gasping is feasible, 53 out of 100 pairs of grasp and hang poses were infeasible for the pick & hang scenario and 86 out of 100 sets for the handover scenario, i.e., many of the sampled poses led to a collision or an infeasible robot configuration for hanging/handover. Some failure cases are depicted in Figs. 18–19. This will become even worse when the whole trajectory is optimized or the mug's initial pose is given arbitrarily. As the sequence length gets longer, not only should an exponentially larger number of planning problems be solved to find a set of feasible poses, but also the found poses are not guaranteed to be optimal. Again, the joint optimization with our constraint models doesn't raise such issues.

### 6.3 6D Pose Estimation and Zero-shot Imitation

Figs. 14 and 15 visualize some principal components of the image features and of the representation vectors, respectively. It can be seen that each component represents a certain property of the objects, such as inside vs. outside, handle vs. other parts, or above vs. below. This enables the image-based pose estimation which we call feature-based closest point (FCP) matching, i.e., the problem of finding the relative pose of a target mesh w.r.t. a model mesh whose pose is given, without defining any canonical coordinate of the objects. To this end, we first queried the backbone at $10^3$ and $5^3$ grid points around the target and the model, respectively, (as shown in Fig. 20(d)) with their own images. For each model grid point, the corresponding target point is obtained such that their representations are closest. Finally, FCP computes the rigid transformation that minimizes the sum of their Euclidean distances. We compared this to the conventional iterative closest point (ICP) algorithm on point clouds, i.e., the problem of finding the pose minimizing the Euclidean distance of two sets of point clouds obtained from depth cameras. Instead of using depth images, the point clouds can be obtained from the reconstructed meshed via the learned SDF features and we call it ICP2. The point clouds' size was 1000. Fig. 6 shows the position and orientation errors of the 6D pose estimation. FCP performs much better especially in the orientation because, as already widely known, ICP easily gets stuck at the local optima; ICP2 was similarly as worse as ICP. Specifically, a significant improvement was observed in F+ICP2 where we used the FCP results as starting points of ICP2 (which is performed without depth images). Fig. 21 depicts some of the results.

Another observation from the PCAs is that the semantics of the representation are consistent across different objects, e.g. the handle parts of different mugs have similar representations, which implies a pose of one object can be transferred into another through it. We therefore considered an image-based zero-shot imitation scenario, where the environment contains one robot arm, one target mug (filled with small balls) and 4 cameras as shown in Fig. 7. We manually designed the pouring motion and captured the camera images of pre- and post-pouring postures of the mug, $\mathcal{P}_{\text{pre}} = (\mathcal{I}_{\text{pre}}, \boldsymbol{T}_{\text{pre}}, \boldsymbol{K}_{\text{pre}})$ and $\mathcal{P}_{\text{post}} = (\mathcal{I}_{\text{post}}, \boldsymbol{T}_{\text{post}}, \boldsymbol{K}_{\text{post}})$, respectively. For a new mug, we solved the sequential manipulation planning with [(GRASP, gripper, mug), (POSEFCP, $\mathcal{P}_{\text{pre}}$, mug), (POSEFCP, $\mathcal{P}_{\text{post}}$, mug)], where (POSEFCP, ·, ·) imposes the aforementioned FCP constraint. That is, the trajectory optimizer tries to match each part of the new object to the corresponding part of the target while coordinating the global consistency of the full trajectory, thereby resulting in the imitation of the reference motions *only* from the posed images. Fig. 7 shows the optimized post-pouring posture; Figs. 22 – 23 and the videos [playlist3] show the reference and its imitations more clearly.

## 7 Conclusion

This work transferred the idea of neural implicit representations to robotic manipulation applications. The pixel-aligned nature enables the proposed representations to be inferred from posed camera images and easily capture fine-grained details of the objects. We demonstrated that the learned representations and interaction features allowed for formulating and solving general sequential manipulation problems only from images within the LGP framework. The learned representations generalize to unseen objects, enabling manipulation planning and zero-shot imitations for them.

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

# A  Appendix

## A.1  Homography Transformation

The idea of the Homography warping is that two images taken by cameras at the same position but with different orientations and intrinsics can be transformed into each other. Suppose that we have a source image $\mathcal{I}$ with the camera position $\boldsymbol{t}$, rotation matrix $\boldsymbol{R}$ and projection matrix $\boldsymbol{K}$ and that an object is inside a bounding sphere at $\boldsymbol{p} \in \mathbb{R}^3$ with a radius $r \in \mathbb{R}^+$. An image focusing on the bounding sphere can be taken from a (synthetic) camera at the same position $\boldsymbol{t}$ with the view direction as $\boldsymbol{t} - \boldsymbol{p}$ and the field of view angle as $2 \arcsin(||\boldsymbol{t} - \boldsymbol{p}||_2 / r)$, from which we can compute the new camera rotation matrix $\hat{\boldsymbol{R}}$ and the intrinsic $\hat{\boldsymbol{K}}$.

Given $\hat{\boldsymbol{R}}$ and $\hat{\boldsymbol{K}}$, the new field warped by the corresponding Homography can be obtained as follows: First, a pixel in the source image, $p_1 = (u, v, 1)$, is reprojected into a ray in the 3D space: $P_1 = \boldsymbol{K}^{-1} p_1$. Next, the ray is viewed in the new camera coordinate: $P_2 = \hat{\boldsymbol{R}}^T \boldsymbol{R} P_1$. Lastly, this ray is projected back into a pixel in the new camera: $p_2 = \hat{\boldsymbol{K}} P_2$. Putting all together, the Homography warping is given as:

$$\mathcal{W}(\hat{\boldsymbol{R}}, \hat{\boldsymbol{K}}) : \begin{bmatrix} u \\ v \\ 1 \end{bmatrix} \mapsto w \hat{\boldsymbol{K}} \hat{\boldsymbol{R}}^T \boldsymbol{R} \boldsymbol{K}^{-1} \begin{bmatrix} u \\ v \\ 1 \end{bmatrix}, \tag{13}$$

where $w$ is the parameter that makes the last element of the output homogeneous coordinate 1, which results in the warped image $\hat{\mathcal{I}}$ with its camera pose $\hat{\boldsymbol{T}} = \begin{bmatrix} \hat{\boldsymbol{R}} & \boldsymbol{t} \\ \boldsymbol{0} & 1 \end{bmatrix}$ and intrinsic matrix $\hat{\boldsymbol{K}}$.

## A.2  Manipulation Constraints

In this work, we consider two discrete actions, (GRASP, gripper, mug) and (HANG, hook, mug), for grasping and hanging, respectively. Each action imposes three constraints on the path as follows.

- The action $a_k = (\text{GRASP}, \text{gripper}, \text{mug})$ first imposes the learned grasping constraint at the end of its phase, $H_{\text{grasp}}^i(\boldsymbol{x}_t, \delta \boldsymbol{q}_t^i) = 0$, $t = kT$, i.e.,

$$\left( T(\delta \boldsymbol{q}_t^i)[\phi_{\text{grasp}}^i] \circ FK_j \right)(\boldsymbol{x}_t) = 0, \tag{14}$$

  where $i$ and $j$ are indices of the mug and the gripper, respectively. It also imposes the zero-impact switching constraint at $t = kT$, i.e.,

$$\hat{\boldsymbol{v}}_t = 0, \tag{15}$$

  where $\hat{\boldsymbol{v}}_t$ is a joint velocity computed from $\boldsymbol{x}_{t-1}$ and $\boldsymbol{x}_t$ via finite difference. Lastly, it introduces an equality constraint on the gripper's approaching direction for collision-free grasping; more precisely, the constraint is imposed at $t \in \{kT - 2, kT - 1, kT\}$ as:

$$^j \hat{\boldsymbol{a}}_t^i = a_{\text{approach}} \begin{bmatrix} 0 \\ 0 \\ -1 \end{bmatrix}, \tag{16}$$

  where $^j \hat{\boldsymbol{a}}_t^i$ is the mug's acceleration in the gripper's coordinate computed from $^j \boldsymbol{t}(\delta \boldsymbol{q}_{t-2}^i)$, $^j \boldsymbol{t}(\delta \boldsymbol{q}_{t-1}^i)$ and $^j \boldsymbol{t}(\delta \boldsymbol{q}_t^i)$ via finite difference, and $a_{\text{approach}} \in \mathbb{R}^+$ is the predefined approaching acceleration magnitude. The gripper's $z$ axis is depicted in Fig. 11(a) as a blue arrow. Combined with the above zero-impact constraint, this constraint enforces the gripper to approach the mug in the gripper's -$z$ axis direction and to stop moving at the end of the phase.

- Similarly, the action $a_k = (\text{HANG}, \text{hook}, \text{mug})$ consists of the learned hanging constraint, the zero-impact and hanging approaching constraints as

$$\left( T(\delta \boldsymbol{q}_{kT}^i)[\phi_{\text{hang}}] \circ FK_j \right)(\boldsymbol{x}_{kT}) = 0, \tag{17}$$

$$\hat{\boldsymbol{v}}_{kT} = 0, \tag{18}$$

$$^j \hat{\boldsymbol{a}}_t^i = a_{\text{approach}} \begin{bmatrix} 0 \\ 0 \\ 1 \end{bmatrix}, \ \forall t \in \{kT - 2, kT - 1, kT\}, \tag{19}$$

  where $i$ and $j$ are indices of the mug and the hook, respectively, and the hook's $z$ axis is the blue arrow in Fig. 11(b) (or outer product of the red and green arrow).

The discrete actions above affect the consecutive symbolic states. While $s_k$ indicates a mug is grasped by a gripper or hung on a hook at the phase $k$, we impose the following path constraint:

$$\delta \boldsymbol{q}_t^i - \delta \boldsymbol{q}_{t-1}^i = FK_j(\boldsymbol{x}_t) - FK_j(\boldsymbol{x}_{t-1}), \ \forall t \in \{(k-1)T+1, \cdots, kT\} \tag{20}$$

where $i$ and $j$ are indices of the mug and the gripper/hook, respectively. Effectively this introduces a static joint between the two frames (Toussaint et al., 2018) so the mug moves along with its parent frame (the gripper or hook). The collision constraints are also imposed along the trajectory, where the pair collisions with the mug are computed by the learned SDF feature. We introduce the collision feature in the following section.

We would like to emphasize that our manipulation planning framework is not limited by the constraints we introduced above, but it can incorporate any existing other constraint models and methods, e.g., (Toussaint et al., 2018; Ha et al., 2020; Toussaint et al., 2020; Driess et al., 2021).

### A.3 DEFINING PAIR-COLLISION CONSTRAINTS WITH SDFs

For manipulation planning problems written only by convex meshes, the distance or penetration of two objects, which we call pair-collision features, are computed with either Gilbert-Johnson-Keerthi (GJK) for non-penetrating objects or Minkowski Portal Refinement (MPR) for penetrating objects. In this section, we introduce how to define pair-collision features when one or both objects are given as SDFs.

**SDF vs. Sphere:** Let $\delta \boldsymbol{q}_i$, $\boldsymbol{q}_j$ and $r_j$ be the rigid transformation of PIFO, the sphere's pose and radius, respectively. Then the pair-collision feature is simply given by:

$$d_{ij} = T(\delta \boldsymbol{q}_i)[\phi_{\text{SDF}}](\boldsymbol{t}(\boldsymbol{q}_j)) - r_j. \tag{21}$$

**SDF vs. Capsule:** Let $\delta \boldsymbol{q}_i$, $\boldsymbol{q}_j$, $h_j$ and $r_j$ be the rigid transformation of PIFO, the capsule's pose, height and radius, respectively. The pair-collision feature is given by the solution of the following optimization:

$$d_{ij} = \min_{-h_j/2 \leq z \leq h_j/2} T(\delta \boldsymbol{q}_i)[\phi_{\text{SDF}}] \left( \boldsymbol{R}(\boldsymbol{q}_j) \begin{bmatrix} 0 \\ 0 \\ z \end{bmatrix} + \boldsymbol{t}(\boldsymbol{q}_j) \right) - r_j. \tag{22}$$

**SDF vs. Mesh:** Let $\delta \boldsymbol{q}_i$ and $\boldsymbol{q}_j$ be the rigid transformation of PIFO, the mesh's pose, respectively.

$$d_{ij} = \min_{\substack{\boldsymbol{p}_1 \in \mathbb{R}^3, \boldsymbol{p}_2 \in \mathbb{R}^3 \\ T(\delta \boldsymbol{q}_i)[\phi_{\text{SDF}}](\boldsymbol{p}_1)=0 \\ d_j(\boldsymbol{p}_2)=0}} \boldsymbol{n}_1^T(\boldsymbol{p}_2 - \boldsymbol{p}_1), \tag{23}$$

where $\boldsymbol{n}_1$ is the normal vector of $\phi_{\text{SDF}}$ at $\boldsymbol{p}_1$ and $d_j(\boldsymbol{p}_2)$ is the signed distance of $\boldsymbol{p}_2$ to the mesh computed by GJK/MPR.

**SDF vs. SDF:** Let $\delta \boldsymbol{q}_i$ and $\delta \boldsymbol{q}_j$ be the rigid transformations of two PIFOs.

$$d_{ij} = \min_{\substack{\boldsymbol{p}_1 \in \mathbb{R}^3, \boldsymbol{p}_2 \in \mathbb{R}^3 \\ T(\delta \boldsymbol{q}_i)[\phi_{\text{SDF}}^i](\boldsymbol{p}_1)=0 \\ T(\delta \boldsymbol{q}_j)[\phi_{\text{SDF}}^j](\boldsymbol{p}_2)=0}} \boldsymbol{n}_1^T(\boldsymbol{p}_2 - \boldsymbol{p}_1). \tag{24}$$

The optimizations in Equation 22–Equation 24 should be run multiple times from different initial guesses because the object shape represented as SDF can be non-convex. In practice, we found approximating the meshes by a number of spheres and computing the collision feature much more efficient because querying the network $\phi_{\text{SDF}}$ at multiple points can be done in parallel on GPUs.

### A.4 NETWORK PARAMETERS

**Image encoder** has the U-net architecture (Ronneberger et al., 2015), especially with the headless ResNet-34 (He et al., 2016) as its downward path and two residual $3 \times 3$ convolutions followed by up-convolution as the upward path. The number of output channels is $64$.

**3D reprojector** computes the coordinate feature as 32-dimensional vector using one linear+ReLU layer and concatenate it with the local image feature. They are passed to two hidden layers with the

width of (256, 128) followed by ReLUs. Therefore, the dimension of the representation vector is 128.

**SDF head** takes as input one representation vector and computes the output through one hidden layer with the width of 128 followed by ReLU.

**Grasp and hang heads** take as input 27 and 5 representation vectors at their interaction points (depicted in Fig. 11) and predict the feature through two hidden layers with the widths of (256, 128) followed by ReLUs.

As shown in Figure 13, the network structures for comparison in Section 6.1 was kept similar to the above as possible. Image encoders of the global image feature and vector representation networks are the ResNet-34 returning 64-dimensional vector. The feature head structures remain the same, but, because the vector representation scheme doesn't represent objects as implicit functions, the input of their feature head is the frame's pose as 7-dimensional vector (3D translation+ 4D quaternion). The grasp and hang heads of the SDF representation scheme take as input 27- and 5-dimensional vectors of their interaction points SDF values.

| # of views | Method | SDF error ($\times 10^{-3}$) | Volumetric IoU | Chamfer-$L_1$ ($\times 10^{-3}$) |
|---|---|---|---|---|
| 2 | PIFO | 2.91 / 4.63 | 0.760 / 0.577 | 6.14 / 8.84 |
| | Global Image Feature | 3.58 / 4.96 | 0.642 / 0.515 | 8.50 / 10.8 |
| | Vector Representation | 15.6 / 15.8 | 0.045 / 0.046 | 39.1 / 40.4 |
| | SDF Representation | **2.11 / 3.48** | **0.786 / 0.622** | **5.78 / 8.13** |
| 4 | PIFO | 2.20 / 3.38 | 0.816 / 0.656 | 5.26 / 6.90 |
| | Global Image Feature | 2.82 / 3.93 | 0.697 / 0.581 | 7.42 / 9.49 |
| | Vector Representation | 15.0 / 15.2 | 0.036 / 0.014 | 38.6 / 39.7 |
| | SDF Representation | **1.43 / 2.73** | **0.845 / 0.667** | **4.90 / 6.83** |
| 8 | PIFO | 1.68 / 2.72 | 0.851 / 0.683 | 4.78 / 6.34 |
| | Global Image Feature | 2.31 / 3.51 | 0.728 / 0.607 | 6.75 / 8.80 |
| | Vector Representation | 14.6 / 15.3 | 0.033 / 0.006 | 38.7 / 40.6 |
| | SDF Representation | **1.07 / 2.07** | **0.878 / 0.703** | **4.51 / 6.06** |

Table 2: SDF Feature Evaluation (Training / Test). The SDF errors were also measured at the same grid points as IoU.

| # of views | Method | Grasp (%) | Grasp+c (%) | Hang (%) | Hang+c (%) |
|---|---|---|---|---|---|
| 2 | PIFO | 65.8 / 55.4 | **82.9 / 77.1** | 87.2 / **71.4** | **88.2 / 72.1** |
|   | Global Image Feature | **67.6 / 63.9** | 80.9 / 70.4 | **88.3** / 70.4 | 86.3 / 71.8 |
|   | Vector Representation | 13.2 / 12.9 | 0.8 / 0.4 | 25.6 / 21.8 | 0.0 / 0.0 |
|   | SDF Representation | 41.2 / 55.3 | 49.6/ 45.7 | 2.6 / 1.1 | 3.3 / 2.1 |
| 4 | PIFO | **69.0 / 63.9** | **88.1 / 82.5** | 88.7 / 75.4 | **94.0 / 78.9** |
|   | Global Image Feature | 62.3 / 61.8 | 82.7 / 75.7 | **90.3 / 75.7** | 91.2 / 78.2 |
|   | Vector Representation | 21.2 / 22.5 | 0.5 / 0.4 | 55.1 / 46.4 | 0.0 / 0.0 |
|   | SDF Representation | 49.1 / 46.1 | 67.9 / 64.3 | 3.3 / 2.9 | 3.7 / 4.3 |
| 8 | PIFO | **71.9 / 69.3** | **88.7 / 85.0** | **91.7 / 80.4** | **96.5 / 82.5** |
|   | Global Image Feature | 71.3 / 67.1 | 84.0 / 79.3 | 91.3 / 77.5 | 92.9 / 80.4 |
|   | Vector Representation | 29.0 / 23.9 | 0.5 / 0.7 | 65.9 / 49.6 | 0.0 / 0.0 |
|   | SDF Representation | 51.4 / 52.1 | 75.5 / 70.4 | 4.6 / 6.1 | 6.3/ 5.7 |

Table 3: Task Feature Evaluation (Training / Test).

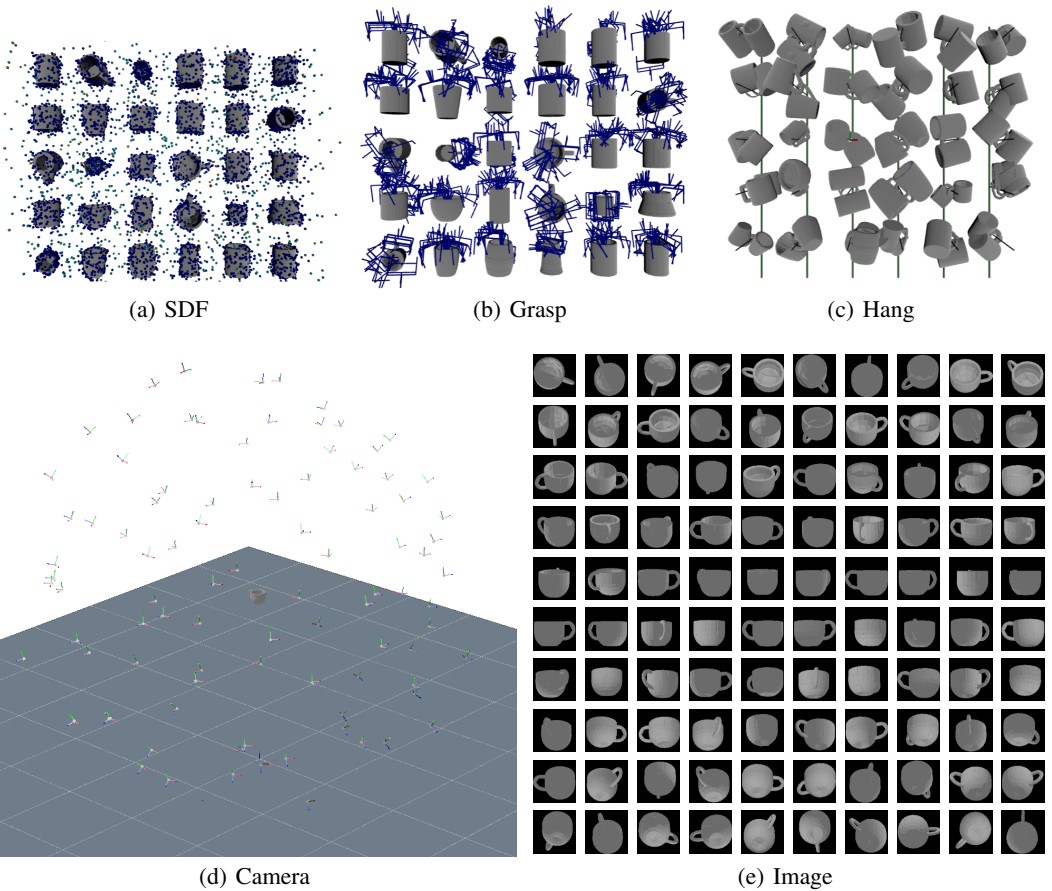

(a) SDF    (b) Grasp    (c) Hang

(d) Camera    (e) Image

Figure 8: Data Generation

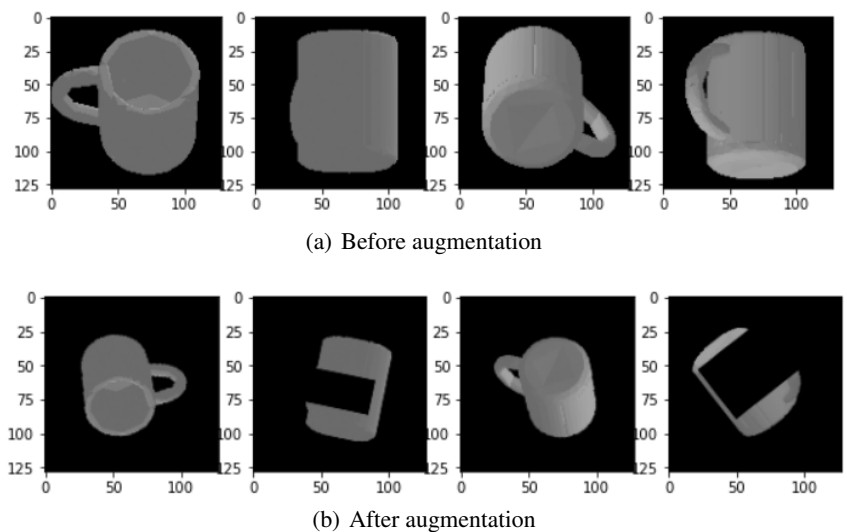

(a) Before augmentation

(b) After augmentation

Figure 9: Image Data Augmentation

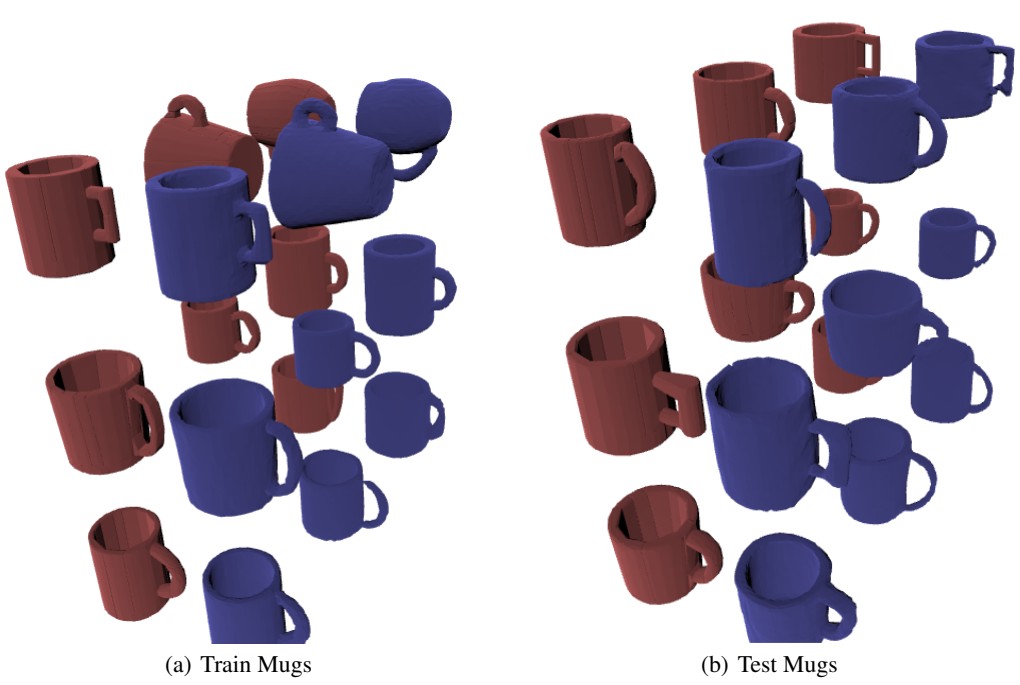

(a) Train Mugs

(b) Test Mugs

Figure 10: Reconstruction via marching cube. Red: ground truth, Blue: reconstructed

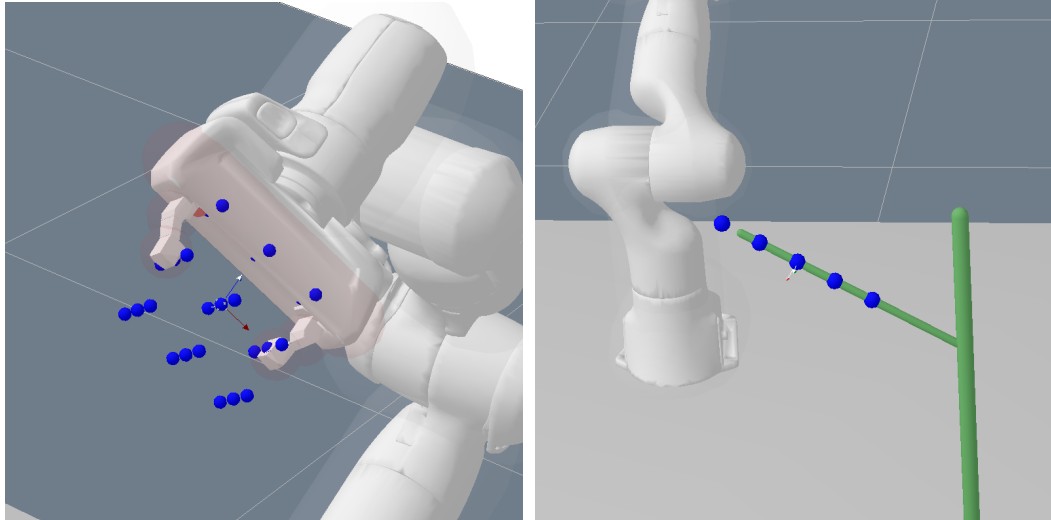

Figure 11: Key interaction points on the gripper and hook. Before passed to PIFO, their global positions are computed from the gripper's or hook's pose $q$, i.e., $p_i = R(q)\hat{p}_i + t(q)$, $\forall i \in \{1, ..., N_{\text{keypoint}}\}$.

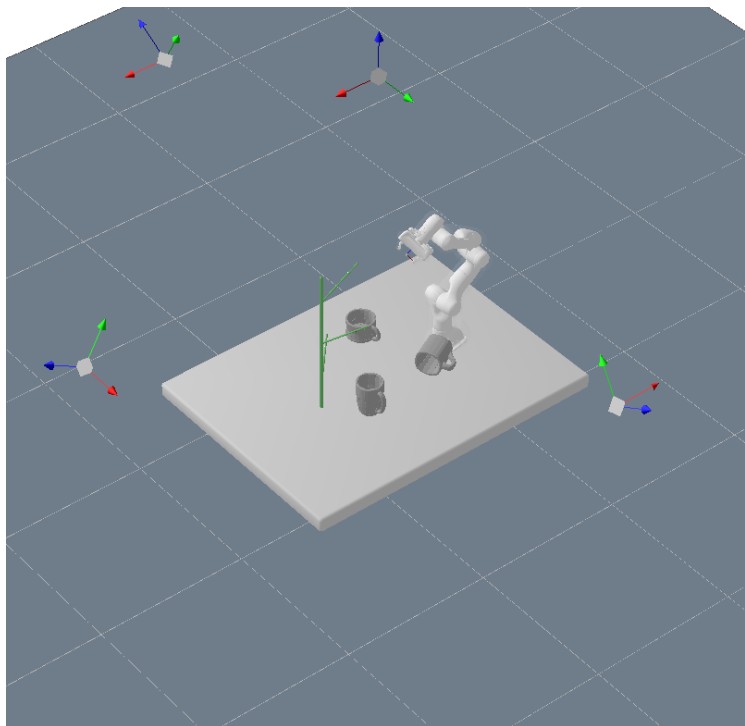

(a) Scene. Four cameras' poses are depicted as coordinate systems where the origin is the camera location, $-z$ axis (blue) is pointing the view direction, and $x$ and $-y$ axes (red and blue) are the directions of $(u, v)$ coordinate of images.

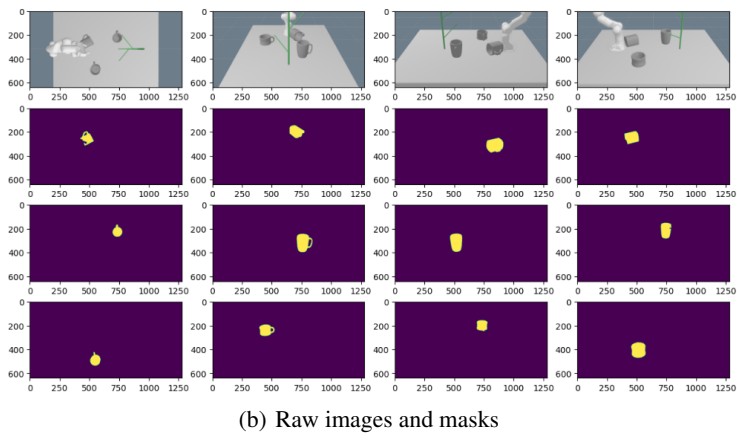

(b) Raw images and masks

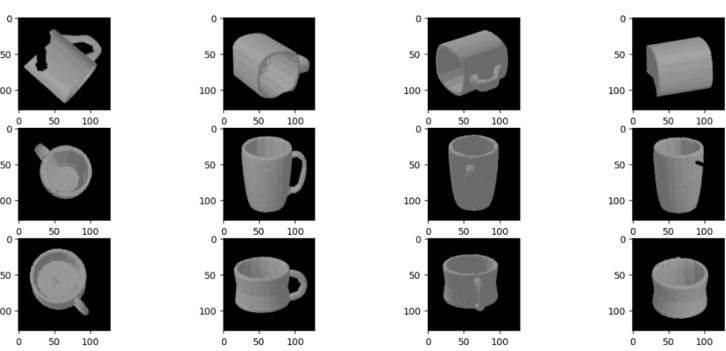

(c) Warped images (via the multi-view processing)

Figure 12: Multi-view processing.

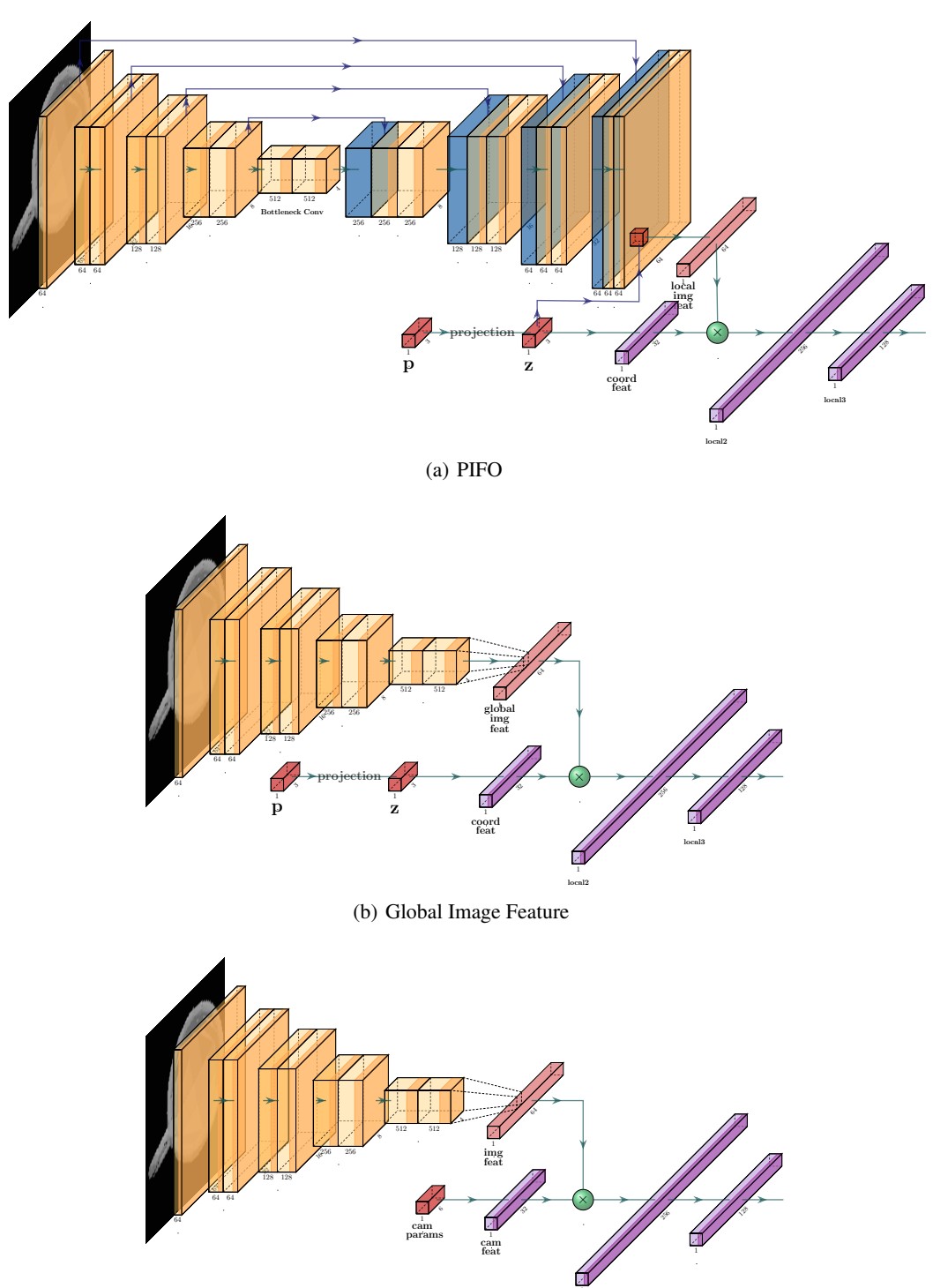

(a) PIFO

(b) Global Image Feature

(c) Vector Object Represnetation

Figure 13: Baseline Networks used for comparison.

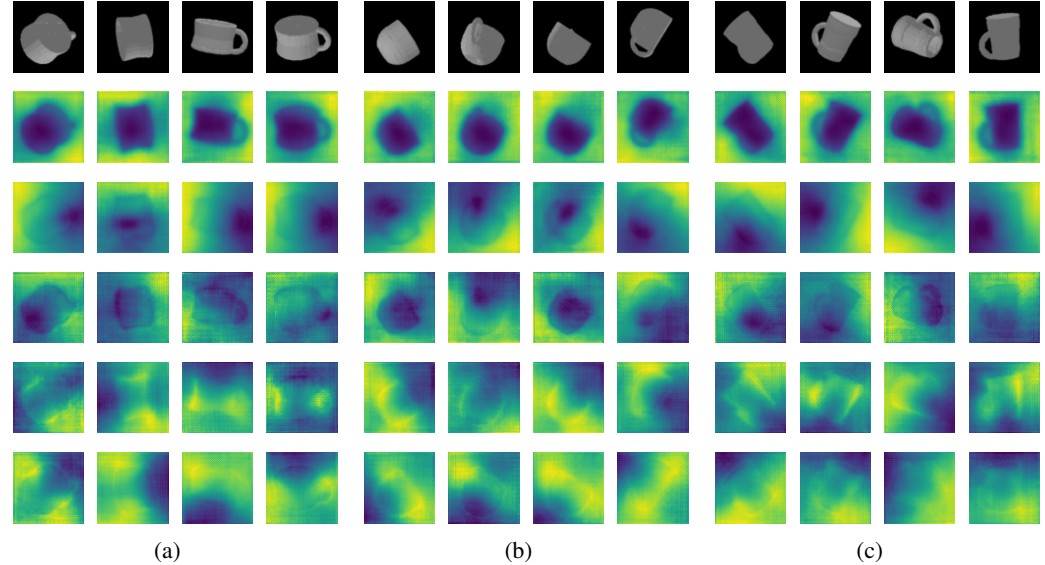

Figure 14: First 5 principal components from PCA on image features. The first component indicates the object vs. non-object areas, the second component distinguishes the handle parts, and the third one spots the above vs. below of the mugs, etc. Note that the network is trained only via the task feature supervisions.

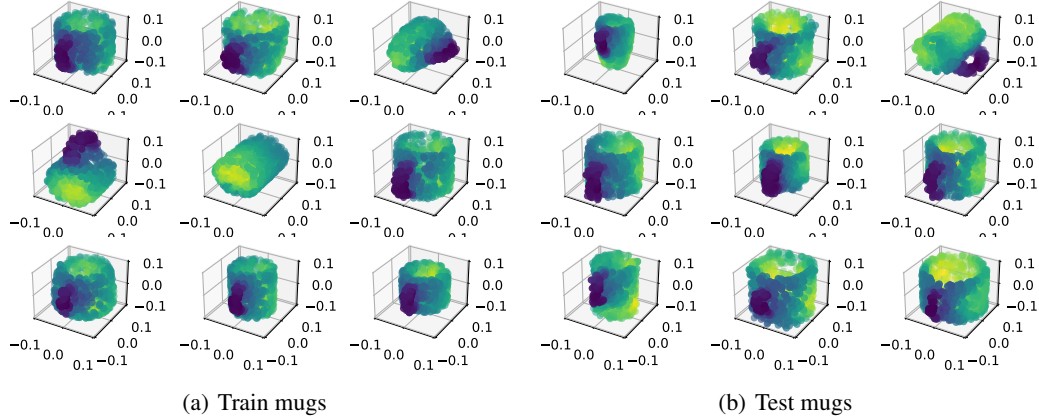

Figure 15: The first principal component from PCA on representation vectors of the 3D surface points. It distinguishes the handles of the mugs from the other parts and is consistent across different mugs.

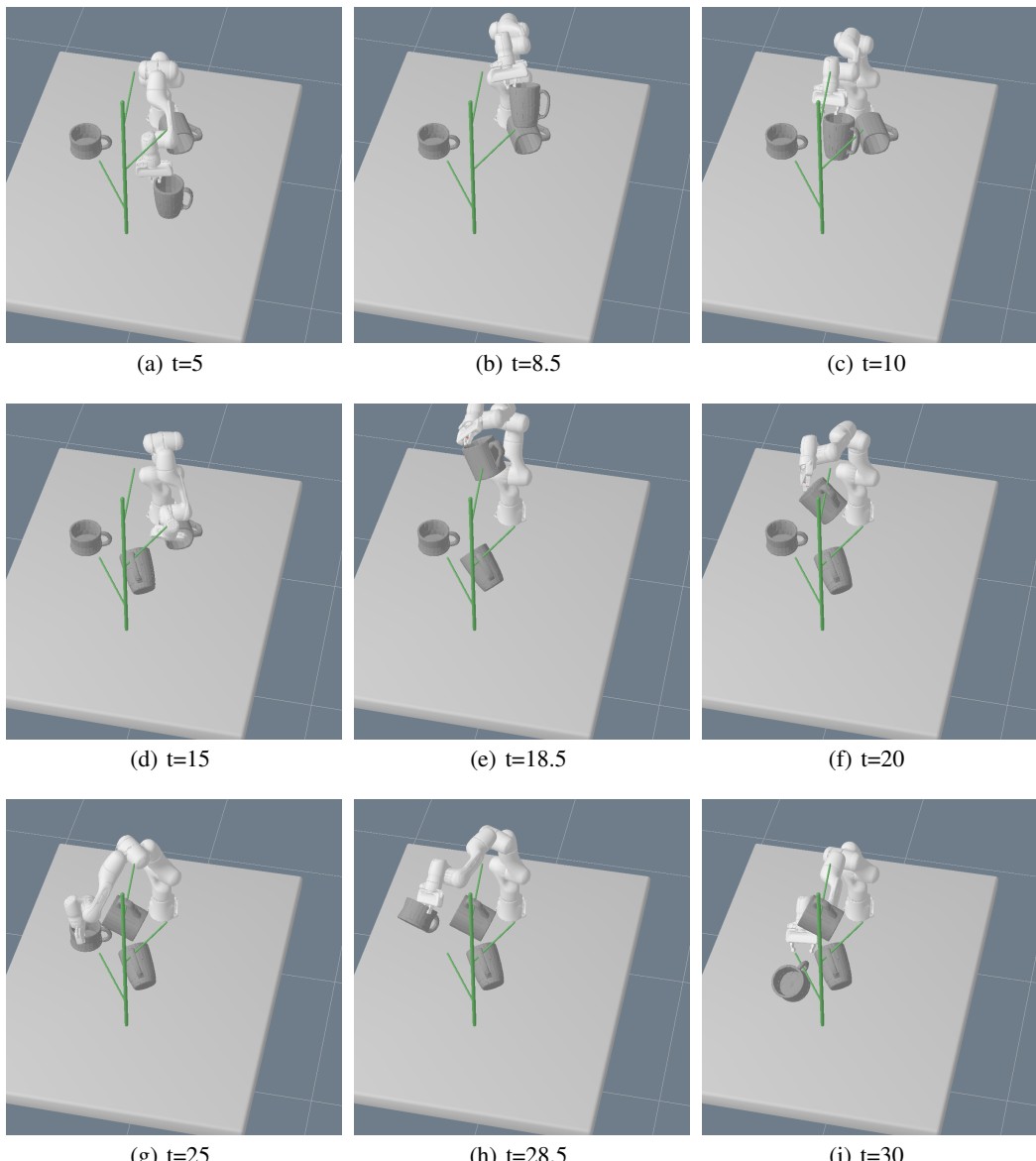

(a) t=5  (b) t=8.5  (c) t=10

(d) t=15  (e) t=18.5  (f) t=20

(g) t=25  (h) t=28.5  (i) t=30

Figure 16: The three-mug scenario. 60 steps of robot configurations and rigid transformations of three mugs are jointly optimized via the proposed manipulation framework. This optimization is a 1071-dimensional decision problem (one 7DOF arm for 60 steps and one 7DOF mug for 51, 31, 11 steps = 1071, the mug's rigid transformations before grasped are not included in optimization) and is solved within 1 minute on a standard laptop.

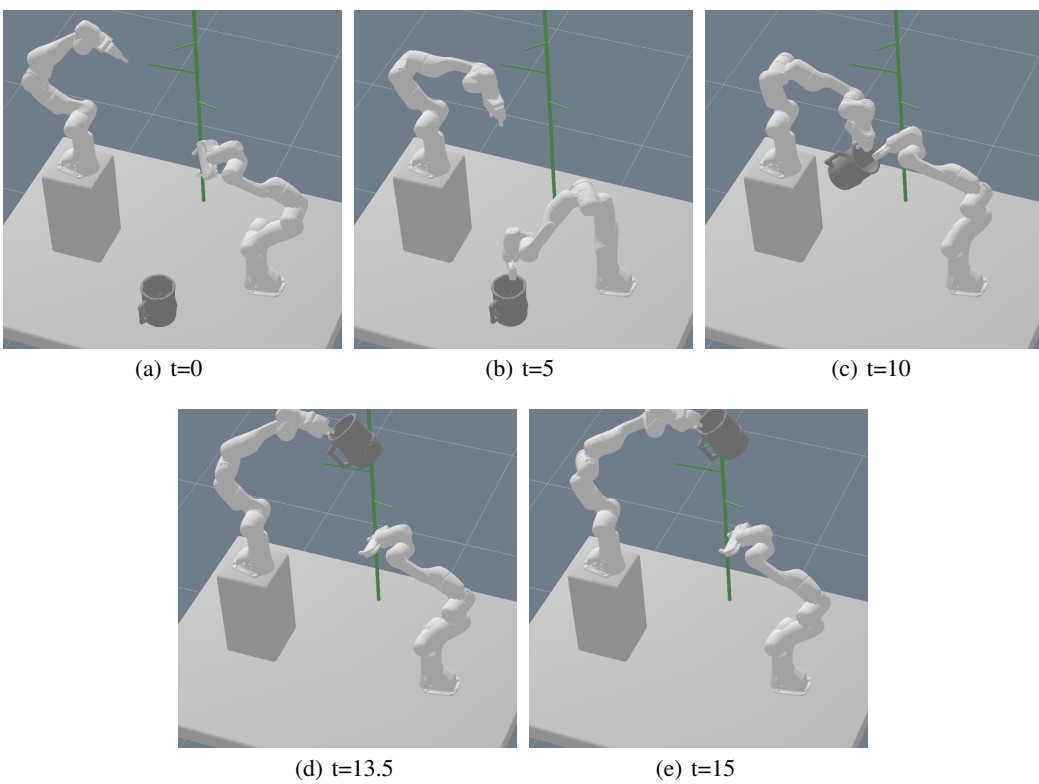

(a) t=0      (b) t=5      (c) t=10

(d) t=13.5     (e) t=15

Figure 17: The handover scenario. 30 steps of the two arms' configurations and rigid transformations of the mug are jointly optimize dvia the proposed manipulation framework. This optimization is a 567-dimensional decision problem (two 7DOF arms for 30 steps and one 7DOF mug for 21 steps = 567, the mug's rigid transformations at the first phase are not included in optimization) and is solved within 1 minute on a standard laptop.

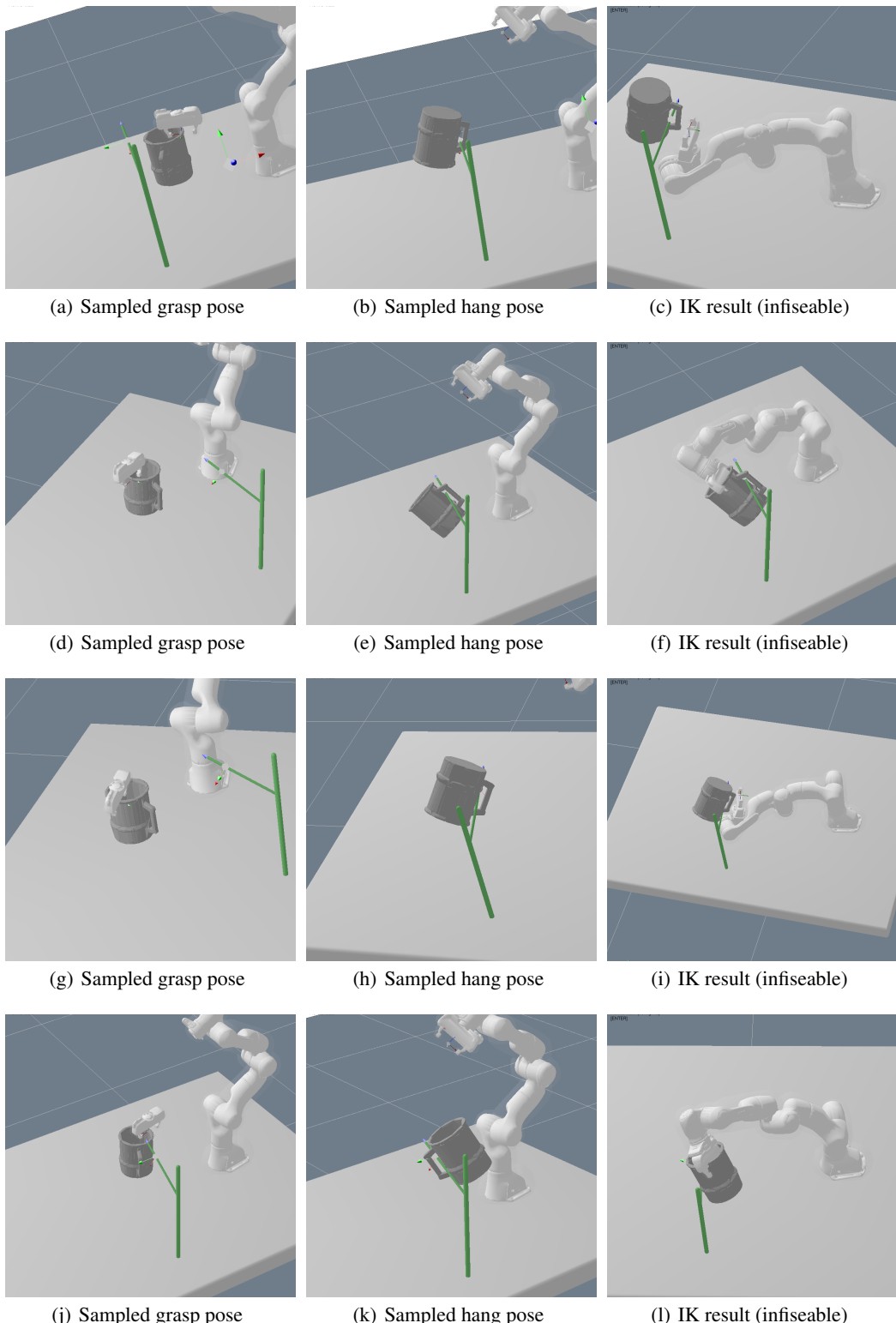

Figure 18: IK with generative models - Pick & Hang. Separately generated poses often can not be coordinated due to the kinematic infeasibility, i.e., the robot joint angle limits, or the collision constraints.

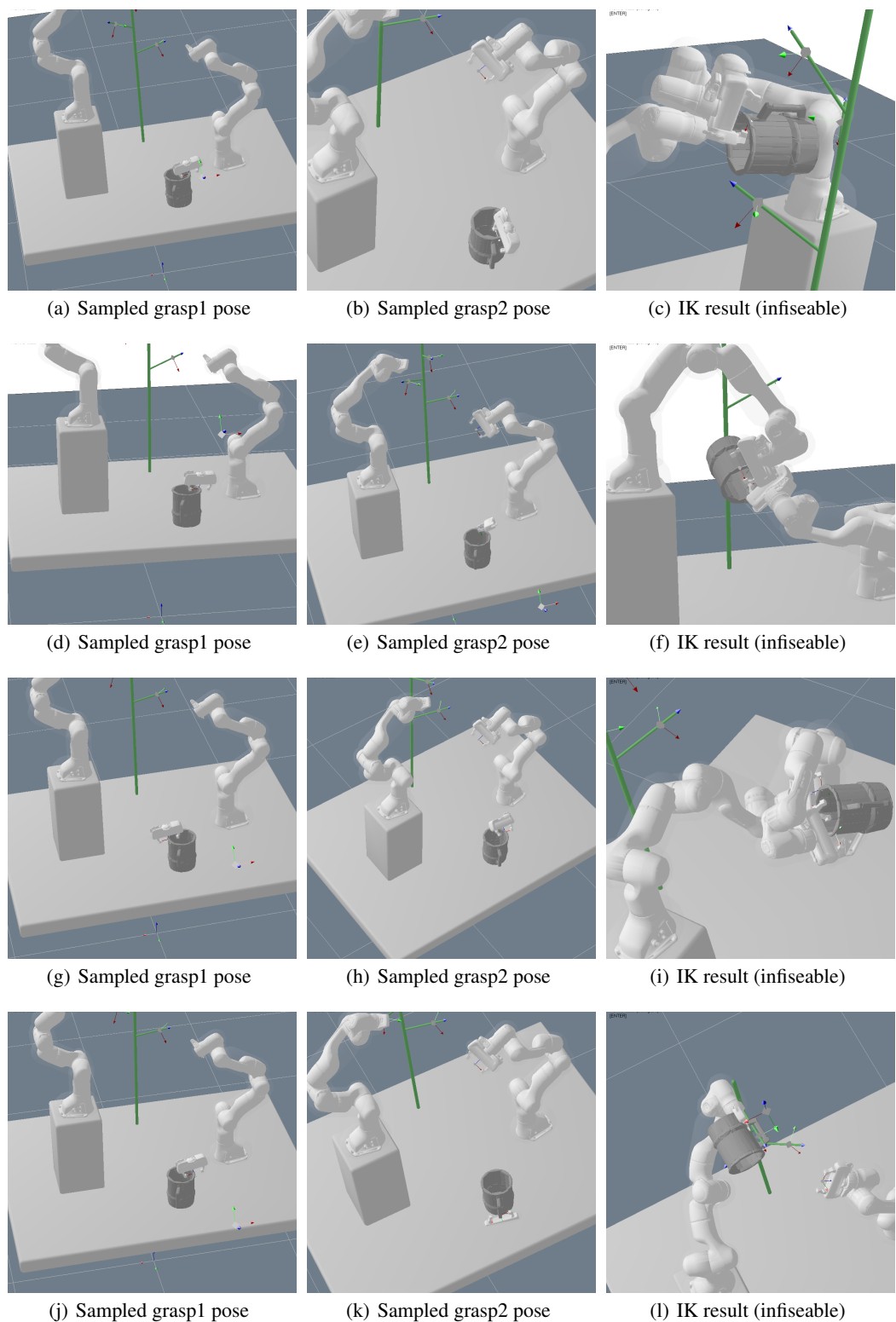

Figure 19: IK with generative models - Handover. Separately generated poses often can not be coordinated due to the kinematic infeasibility, i.e., the robot joint angle limits, or the collision constraints.

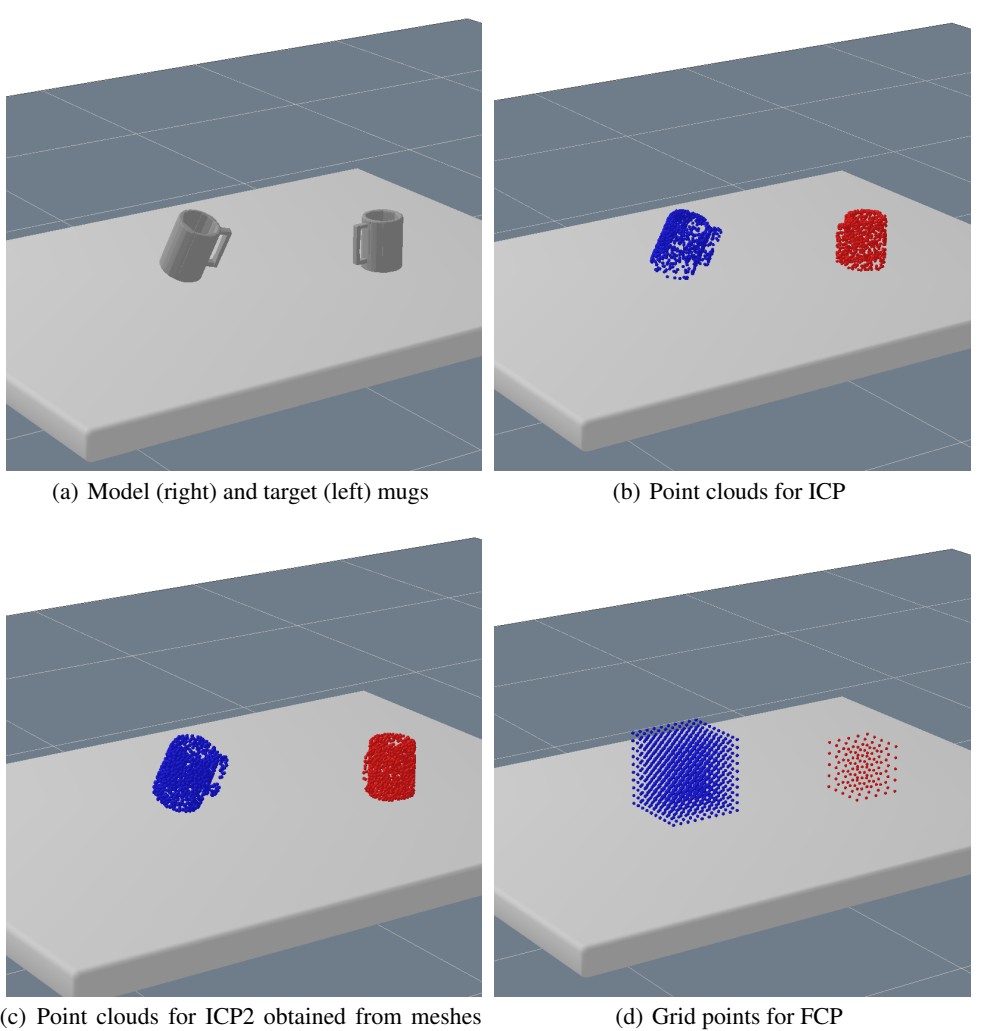

(a) Model (right) and target (left) mugs

(b) Point clouds for ICP

(c) Point clouds for ICP2 obtained from meshes reconstructed via $\phi_{\text{SDF}}$

(d) Grid points for FCP

Figure 20: 6D Pose Estimation. (b) Point clouds for ICP are obtained from depth cameras at the same locations/orientations as the RGB cameras. The size of the point clouds is 1000. (c) Point clouds for ICP are sampled from the surfaces of the meshes reconstructed via the learned $\phi_{\text{SDF}}$. The size of the point clouds is 1000. (d) FCP uses $10^3$ grid points for the target and $5^3$ grid points (in smaller area) for the model, respectively.

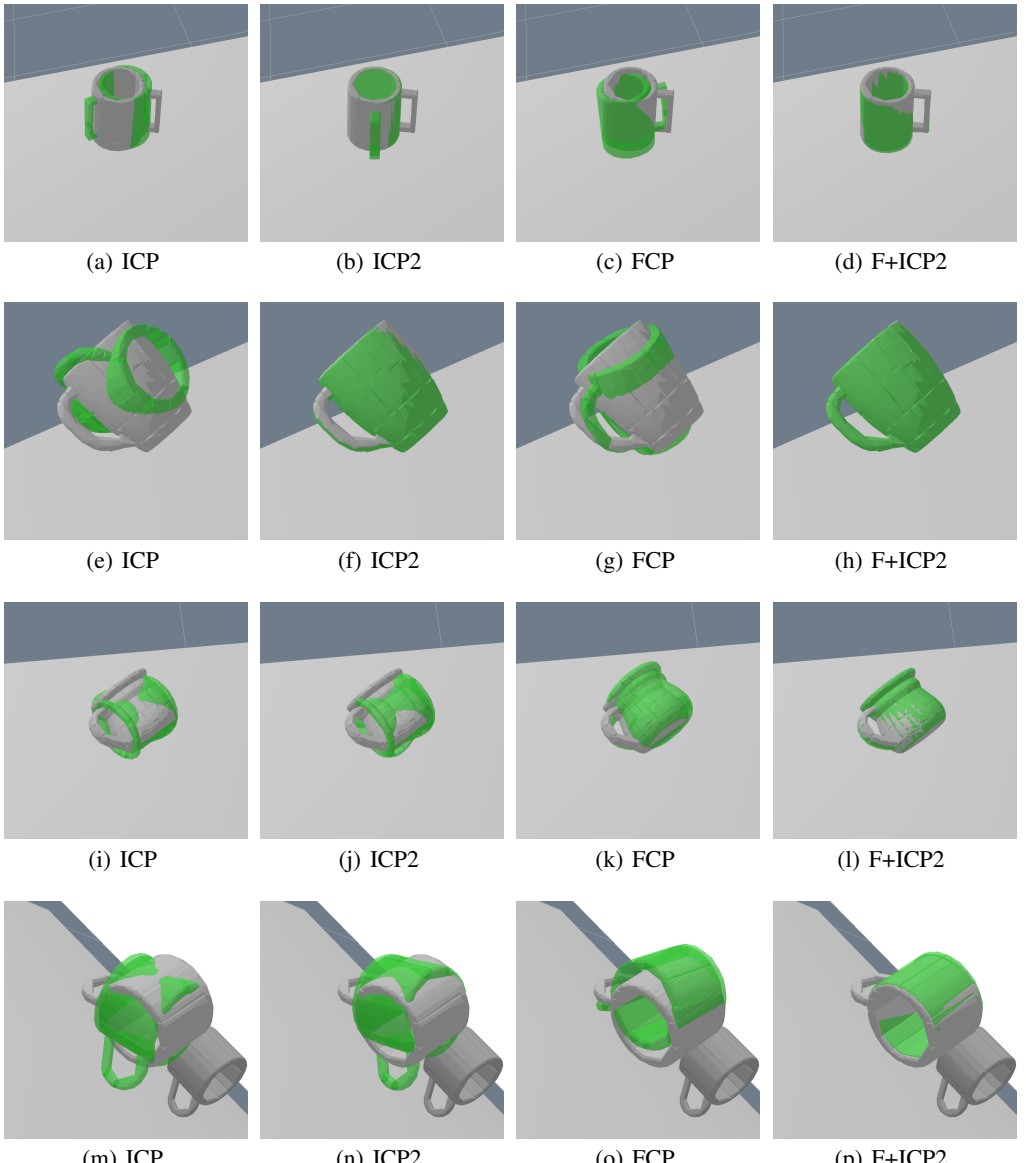

Figure 21: 6D Pose Estimation Results - the estimated poses are applied to the green meshes. ICP easily gets stuck at local optima while FCP produces fairly accurate poses which help F+ICP2 escape the local optima; note that FCP does not iterate to get the results.

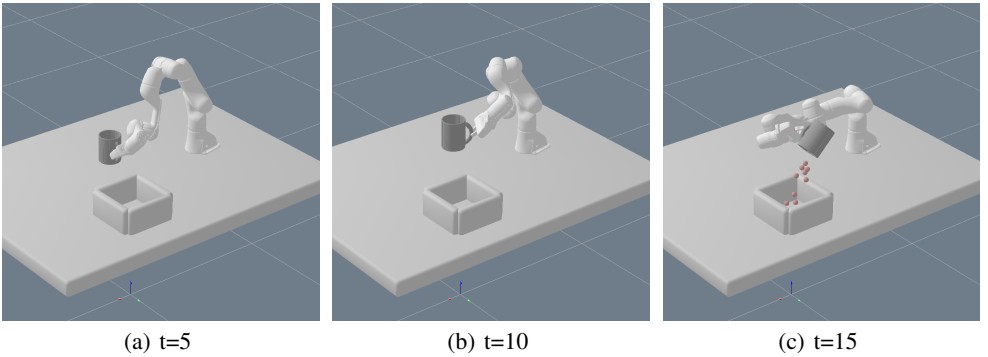

(a) t=5                (b) t=10                (c) t=15

Figure 22: Zero-shot Imitation - reference motion. Two sets of posed images are obtained at $t = 10, 15$.

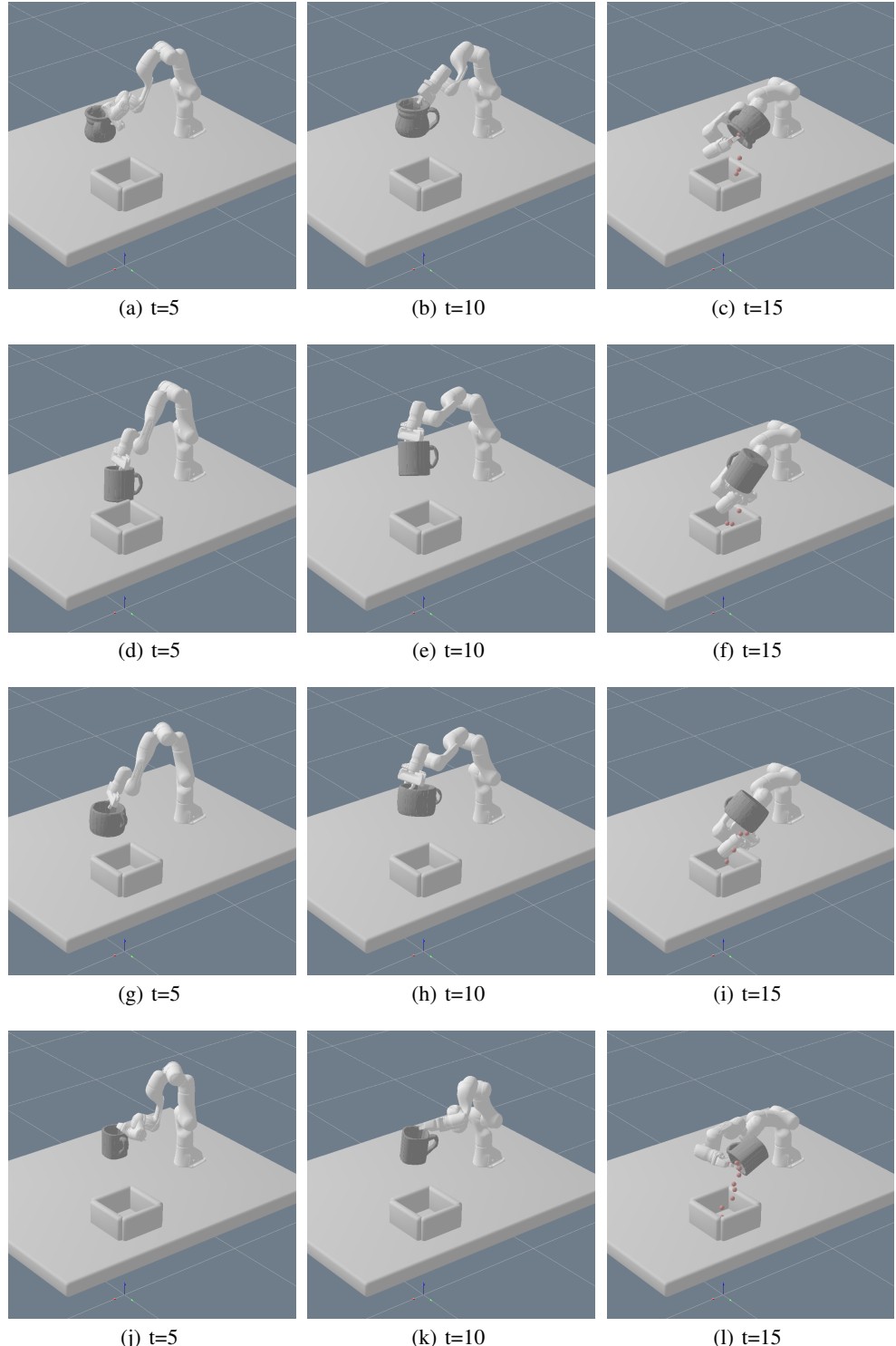

(a) t=5  (b) t=10  (c) t=15

(d) t=5  (e) t=10  (f) t=15

(g) t=5  (h) t=10  (i) t=15

(j) t=5  (k) t=10  (l) t=15

Figure 23: Zero-shot imitation - optimized motions. The FCP constraints are imposed at $t = 10,\ 15$. The imitations are achieved only from images, without defining the canonical coordinate/pose of the objects.

