# OpenReview forum: "Learning Neural Implicit Functions as Object Representations for Robotic Manipulation"
_ICLR.cc/2022/Conference — ICLR 2022 Submitted_

### Official Review · Reviewer_23pA · 2021-10-18

**Correctness:** 3
**Technical Novelty And Significance:** 3
**Empirical Novelty And Significance:** 3
**Recommendation:** 6
**Confidence:** 4

**Main Review:**

Strengths:
- Neural implicit functions are still quite new, and investigating their use for robotic manipulation is a valuable direction. In particular, I think the idea of incorporating NIFs with LGP is worth exploring.

Weaknesses:
- There are not enough quantitative results in the paper to convince the reader that this method is better than others. Also, hanging mugs only is a relatively narrow set of experiments (as the paper is titled "... for robotic manipulation").
- There are no comparisons to other works. Since the introduction mentions that traditional methods (that use traditional representations such as meshes or point clouds) has limitations, a comparison should be made with such a method. Further comparisons should include the Omnihang paper (You et al.). Additionally, the ablations are not that meaningful. It is well known that dense features will give better results than a single global vector (or a single-vector representation of an object).
- The exposition could be more clear. For example, PIFO predicts graspness scores and hang-ness scores, however this is not mentioned in the introduction. The introduction is rather vague. Additionally, it is not clear how exactly the NIF is used in Equation 4. This seems to be a major contribution of the paper, yet it lacks detail.


Questions/Comments:
- In P1 of Section 1, a stated problem with traditional approaches is that "the representations have to be inferred from raw sensory inputs like images or point clouds", however this contradicts the fact that the proposed method infers the NIF from images. Additionally, the last line of P2 also directly contradicts this.
- "Interaction features" should be defined somewhere, it is not clear what this means in the introduction.
- The introduction is lacking citations. For example, NeRF (Mildenhall et al.) should be cited when discussing it.
- In Section 3.5, there has been no discussion of data augmentation up to this point. This is confusing, please clarify this.
- In Eq. 6, what is w? This is not explained in the main paper nor the appendix.
- It is unclear how the graps/hang interaction points of the gripper/hook are used with the neural implicit function. How is the implicit function used to evaluate a grasp or a hang? Please clarify this.
- In 5.1, the ablations (global latent, vector repr, SDF repr) have not yet been introduced. Please re-arrange sections for better flow of the text.
- Some graphics of the baseline networks in the appendix would be helpful for the reader to better understand the baselines and what is being ablated.
- The handover experiment for hanging is complex and interesting. It would be interesting to expand on those experiments and show that other baseline methods cannot solve such a complex task.
- typos:
    - P2 of 2.2: "trowing" -> "throwing"
    - Eq. 5: I believe it should be Mn(u,v)=1
    - P2 of 5.1: "interaction" -> "intersection"
- Incorrect Citations
    - Barron et al. (MIP-NeRF) is ICCV, not CVPR.
    - Trevithick & Yang (GRF) is ICCV, not CVPR.
    - Jiang et al. (Synergies Between...) is RSS, not ICRA.
    - I would suggest double checking all citations.
- Missing related works:
    - Henzler et al. "Unsupervised Learning of 3D Object Categories from Videos in the Wild", CVPR 2021.
    - Reizenstein et al. "Common Objects in 3D: Large-Scale Learning and Evaluation of Real-life 3D Category Reconstruction", ICCV 2021.
    - Van der Merwe et al. "Learning Continuous 3D Reconstructions for Geometrically Aware Grasping", ICRA 2020.

**Summary Of The Paper:**

This paper proposes a method that integrates neural implicit functions (NIFs) with planning methods for robot manipulation. A neural implicit function that represents geometry with SDFs, grasp scores, and hanging scores is learned. This is then integrated with a planner based on Logic-Geometric Planning (LGP). Experiments are run to test mug hanging in a few different scenarios, including hanging a single mug and hanging a mug with handover.


**Summary Of The Review:**

While I like the direction the paper is investigating, I do not think the work is mature enough yet to warrant publication at ICLR. More experiments, comparisons, and clearer exposition would definitely make this paper a better candidate for publication.

---- UPDATE -----
Raised score from 3 to 5.

---- UPDATE -----
Raised score from 5 to 6 in light of comparison to hand-engineered methods.

---

> ### Author Response · Authors · 2021-11-22
> **Thanks a lot for your feedback. We have revised the manuscript and added more experiments**
>
> Thanks a lot for your feedback!
>
> > [W1, 2, Q9] There are not enough quantitative results in the paper to convince the reader that this method is better than others. Also, hanging mugs only is a relatively narrow set of experiments (as the paper is titled "... for robotic manipulation"). ... There are no comparisons to other works. Since the introduction mentions that traditional methods (that use traditional representations such as meshes or point clouds) has limitations, a comparison should be made with such a method. Further comparisons should include the Omnihang paper (You et al.). Additionally, the ablations are not that meaningful. It is well known that dense features will give better results than a single global vector (or a single-vector representation of an object). ... The handover experiment for hanging is complex and interesting. It would be interesting to expand on those experiments and show that other baseline methods cannot solve such a complex task.
>
> The unique feature of this work is that we train interactions as constraint models which can be integrated into general sequential manipulation planning frameworks. Although we trained only SDF, grasping, and hanging constraints, various scenarios from basic pick&hang to handover can be addressed by combining those constraints differently and the proposed framework is general enough to be expanded to other interactions without any modifications.
>
> Most existing works generate each interaction pose individually and later combine them, which is problematic for long-horizon sequential manipulation planning, since combining individual samples from generative models implies a combinatorial complexity. For example, how to grasp the mug should be determined by how to hang it on the hook later, and vice versa. For the handover scenario, grasping, handover, and hanging poses as well as an entire trajectory to achieve those poses should be optimized altogether. We conducted additional experiments on inverse kinematics with generative models and found that only 14 out of 100 sets of individually generated poses were kinematically feasible for the handover scenario. The result is added in Section 6.2.
>
> In the same sense, Omnihang is not subject to comparison because their framework only generates hanging configurations, not grasping nor more general sequential manipulation.
>
> The revision is also including a comparison of the proposed feature-based pose estimation method vs. the conventional ICP algorithm on point clouds where we found that, as already well-known, ICP easily gets stuck at local optima especially for its orientation estimation and the closest point matching based on the feature distance greatly alleviates the issue. The result can be found in Section 6.3.
> Given that this work firstly proposes the usage of neural implicit representations for modeling manipulation constraints, we believe the ablations in 6.1 are meaningful. The comparison with global image features, vector object representations, and SDF representations highlights the benefits of having dense image features, implicit object representations, and task-guided learning scheme, respectively.
>
> > [W3] The exposition could be more clear. For example, PIFO predicts graspness scores and hang-ness scores, however this is not mentioned in the introduction. The introduction is rather vague. Additionally, it is not clear how exactly the NIF is used in Equation 4 (LGP). This seems to be a major contribution of the paper, yet it lacks detail.
>
> We revised the introduction to expose the ideas/contributions more clearly.
> Since the LGP integration of the learned feature is one of the main contributions of this work as you said, we reformatted the paper as follows: Section 3 is now solely about the task feature prediction (as shown in Figure 1), and Section 5 discusses the actual manipulation planning where we added the overview of how the learned feature is integrated into LGP as well as Figure 3 which shows how the various components come together.
>
> > [Q1] In P1 of Section 1, a stated problem with traditional approaches is that "the representations have to be inferred from raw sensory inputs like images or point clouds", however this contradicts the fact that the proposed method infers the NIF from images. Additionally, the last line of P2 also directly contradicts this.
>
> A crucial difference is that our perception pipeline from images to representations is jointly trained with the other parts by the downstream task supervision so the perception objective is aligned with the task feature prediction. The objective of the traditional perception module is to minimize the reconstruction error of the geometry and/or appearance.

---

> > ### Author Response · Authors · 2021-11-22
> > **continue**
> >
> > > [Q2, 3, 4] “Interaction features" should be defined somewhere, it is not clear what this means in the introduction. The introduction is lacking citations. For example, NeRF (Mildenhall et al.) should be cited when discussing it. In Section 3.5, there has been no discussion of data augmentation up to this point. This is confusing, please clarify this.
> >
> > Thanks for pointing these out. The issues were addressed in the revised manuscript.
> >
> > > [Q5] In Eq. 6, what is w? This is not explained in the main paper nor the appendix.
> >
> > $w$ is a parameter that makes a homogeneous coordinate have a form of  (u,v,1). We clarified it in the appendix.
> >
> > > [Q6] It is unclear how the graps/hang interaction points of the gripper/hook are used with the neural implicit function. How is the implicit function used to evaluate a grasp or a hang? Please clarify this.
> >
> > Figure 11 shows the grasp/hang interaction points. Given a gripper or hook’s pose, we can compute the corresponding interaction points’ positions and the implicit representations are obtained at those points. We believe Figures 1, 3 and the added overviews in the revised paper would clarify this.
> >
> > > [Q7, 8] In 5.1, the ablations (global latent, vector repr, SDF repr) have not yet been introduced. Please re-arrange sections for better flow of the text. Some graphics of the baseline networks in the appendix would be helpful for the reader to better understand the baselines and what is being ablated.
> >
> > Thanks for the suggestion. We rearrange the subsection as baselines, metrics and results and added the graphics of the baseline networks in Figure 13.
> >
> > > [Q10, 11, 12] typos, Incorrect Citations,  Missing related works:
> >
> > Thanks. The issues were addressed in the revision.
> >
> > > **Summary Of The Review**:
> > While I like the direction the paper is investigating, I do not think the work is mature enough yet to warrant publication at ICLR. More experiments, comparisons, and clearer exposition would definitely make this paper a better candidate for publication.
> >
> > Thanks again for your feedback. We made a major revision to restructure the paper. We believe that the revised manuscript exposes the main contributions of this work more clearly.
> >
> > The revision includes two additional experiments on inverse kinematics with generative models and pose estimations of conventional ICP which highlight the advantages of the proposed framework over existing works.

---

> > > ### Comment · Reviewer_23pA · 2021-11-24
> > > **Thank you for the paper revision.**
> > >
> > > Thanks for the paper revision. I have seen that many of my exposition questions have been cleared up, and the paper reads with a much better flow. The experiments have been updated with more comparisons. Additionally, with the re-statement of the contributions, I now see where the authors are making their contributions and the implications of them.
> > >
> > > I am upgrading my score from a 3 (when I thought the paper wasn't mature enough to warrant ICLR acceptance) to a 5 (weak reject). The main reason for not pushing my score above the boundary is that the experiments are still lacking. Here are some details:
> > > - The authors included an experiment for 6D pose estimation using NIF features vs. traditional representations (point cloud). This is an interesting experiment, but it doesn't necessarily support the hypothesis that NIFs are better than traditional representation for manipulation. A better baseline comparison would be using traditional representations and hand-engineered interaction constraints with LGP. Showing that the proposed method works better than such a baseline would clearly support the authors' hypothesis stated in the abstract/into.
> > > - The experiments right now are all conducted on synthetic data. From a perception standpoint, these problems are much easier to solve. Some results on real data would make the paper quite strong.
> > > - The authors mentioned the method can be adapted to other object categories and/or tasks without modification. Thus, it should be not too difficult to carry out such experiments to empirically show the generality of the method.
> > > - Some quantitative results are still missing, including success rates for 3-mug hanging and handover.
> > >
> > > If such improvements are made, I believe the paper will be quite strong for any conference submission in the future.

---

> > > > ### Author Response · Authors · 2021-11-29
> > > > **Thank you a lot for the additional feedback!**
> > > >
> > > > We deeply appreciate your time and effort in this review and also thank you for raising your score! Based on your feedback, we conducted additional experiments on the traditional representations + hand-engineered interaction constraints.
> > > >
> > > > Throughout the experiments, objects are represented by meshes, especially with the convex-decomposition (https://github.com/kmammou/v-hacd) for non-convex shapes, and thus pair-distance and collision between meshes can be computed via the GJK/MPR algorithm. On top of this mesh representation, the grasping and hanging constraints are defined and optimized as follows:
> > > > - The grasping constraint consists of the aforementioned collision constraints and the so-called oppose constraint. The oppose feature takes as input three meshes, finger1, finger2, and (a set of decomposed) object meshes to grasp. It computes the minimum pair-distances from finger1 and finger2 to the object and returns the sum of two vectors, i.e., (finger1->object + finger2->object). Making the oppose feature (0,0,0) places the object in the middle of two fingers with proper orientation. This is a widely used hand-engineered constraint that works very well for simple shapes, such as spheres, capsules, etc. Because the mug shapes are highly non-convex we ran the optimization from 100 initial seeds and took the best one with the minimum constraint violation.
> > > > - Given the object mesh, the hanging feature iteratively generates a collision-free pose (up to 10,000 iterations) and checks if the hook is kinematically trapped by the mug (as done in data generation). If trapped, it returns the pose difference so that optimizer can output the found pose.
> > > >
> > > > These features are evaluated with the meshes reconstructed by the learned SDF representations (4 views) as well as with the ground-truth meshes. The grasping success rates (Training / Test mugs) were **62.8% / 75.0%** on the GT meshes and **66.7% / 42.9%** on the reconstructed meshes. The hanging success rates were **94.9% / 92.9%** and **78.2% / 60.7%** on the reconstructed meshes. The comparisons with other representations can be easily seen in this table [[Link]](https://drive.google.com/file/d/1LYFjxG-BU3wZQWbAjlCKUWYG-jQg4KW1/view?usp=sharing).
> > > >
> > > > As shown in the figures of some failure cases [[Link]](https://drive.google.com/file/d/16cmhJsLiyqnUNBEonEqDnJXP2z__gkGL/view?usp=sharing), the reconstruction error of a mesh is directly associated with the planning result; e.g. it would never grasp not-reconstructed parts, would try to hang the mug through a wrongly-generated hole, and/or could result in collisions. While the perception pipeline for this mesh representation is never encouraged to reconstruct the “graspable/hangable parts” more accurately, we can view our end-to-end training via task supervision as a way to do so. Moreover, the hand-engineered feature sometimes produces a wrong grasping pose even for the ground truth mesh (e.g., (a) in the figure above). One can argue that a better interaction feature could be hand-designed by investigating the physics and kinematic structures more deeply, but that would require a huge amount of human insights/efforts and thus is less scalable. In contrast, our data-driven approach eliminates this procedure and directly learns the constraint features from empirical success data of physical interactions.
> > > >
> > > >
> > > > The idea of working with real images is definitely interesting and worth investigating. This can be done by adopting the sim-to-real transfer techniques [1] or RL-style data collection methods in the real world and there’s whole other literature devoted only to this topic. We left it as future work because we believe that training the image-based constraint features in simulation is non-trivial on its own and provides sufficient contributions as the first step in this direction. Also please note that the evidence that neural implicit functions are capable of dealing with real images is actively being provided in the recent literature.
> > > >
> > > > [1] Peng, Xue Bin, et al. "Sim-to-real transfer of robotic control with dynamics randomization." 2018 IEEE international conference on robotics and automation (ICRA). IEEE, 2018.
> > > >
> > > > Of course, the proposed method is applicable to other object categories by collecting the corresponding training data. As we mentioned already, however, defining/learning object representation and interaction features have been a long-standing problem in robotics and we showed in the paper that the learned features generalize to any objects within the category.

---

> > > > > ### Author Response · Authors · 2021-11-29
> > > > > **continue**
> > > > >
> > > > > Regarding the range of the experiments provided in the paper, the comparisons in section 6.1 (including the newly added ones above) show how well the learned features are aligned with the actual physical feasibility, and the quantitative results on the pick-and-hang in 6.2 exhibits the compatibility of the learned features with the trajectory optimization algorithm. We’ve only provided qualitative demonstrations for long-horizon planning scenarios because the success rate of such scenarios heavily depends on the given symbolic state sequence and in practice, such feasible symbolic states should be predicted by some heuristics [2,3] instead of exhaustively testing all.
> > > > >
> > > > > [2] Driess, Danny, Jung-Su Ha, and Marc Toussaint. "Deep Visual Reasoning: Learning to Predict Action Sequences for Task and Motion Planning from an Initial Scene Image." Robotics: Science and Systems (RSS), 2020.
> > > > >
> > > > > [3] Yuan, Wentao, et al. "SORNet: Spatial Object-Centric Representations for Sequential Manipulation." 5th Annual Conference on Robot Learning. 2021.
> > > > >
> > > > >
> > > > > To summarize our answers, we’ve performed additional experiments on the traditional mesh representations and hand-engineered interaction constraints. We think these new results have justified the necessity and advantage of our data-driven approach and, with other experimental results, the hypotheses made in the paper are now well supported. We also strongly believe that this work is well-aligned with the theme of ICLR, highlighting yet another motivation of learning representations, and is an important improvement in robotics as well. Thank you again for your detailed reviews and, if our answers and the additional experiments have cleared up some of your concerns, please consider reassessing the paper.

---

> > > > > > ### Comment · Reviewer_23pA · 2021-11-29
> > > > > > **Good Comparison to Hand-Engineered Method**
> > > > > >
> > > > > > I thank the authors for providing another experiment that compares a more traditional representation/method with PIFO. This greatly helps in validating the hypothesis. The qualitative result also gives insight into how convex-decomposition of meshes can lead to further issues with manipulation planning for hanging, which is quite interesting (and I believe should be of interest to the community). Please cite some references for the "widely-used hand-engineered constraint".
> > > > > >
> > > > > > With the amount of experiments and insights, I believe the paper has enough novelty both technically and experimentally to push it over the border of acceptance. Additionally, the exposition revision makes the paper much clearer and easier to read. Thus, I am upgrading my score from a 5 (weak reject) to a 6 (weak accept). I still think that with experiments with different object classes and real-data experiments, this paper would become a strong accept.

---

### Official Review · Reviewer_seqp · 2021-10-27

**Correctness:** 3
**Technical Novelty And Significance:** 2
**Empirical Novelty And Significance:** 2
**Recommendation:** 5
**Confidence:** 3

**Main Review:**

Strength:
1. The overall idea of combining pixel-wise image features, the implicit function over the 3D space, and task guided learning scheme makes sense to me. However, the separate idea of each component may not be very novel.

2. The authors present some experiments (with videos) to illustrate their method.

Weakness & confusions:
1. The authors only compared their method with some ablated versions (i.e., global features, SDF representation) and didn't compare with other methods:
    (a) methods also propose object representation for robot manipulation. Besides the works mentioned (but not compared) in related work, SORNet[1] is also a work needed to cite and compare.
    (b) traditional baselines: pose estimation + heuristic landmarks + motion planning + (RL) + (grasp point proposal). There are lots of strong baselines for the pick and place style tasks. What's the advantage over them, and why not compare with them?

2. It's not clear how the learned features are incorporated into the LGP framework. What are the detailed examples of h_path, g_path, h_switch, and g_switch (equation (4)) for the grasp and hang experiments?

3. The current representation only cares about the relationship between the object and the pre-defined key points (e.g., gripper and fixed hook). It doesn't capture the relationship between the objects (e.g., whether object A is on top of object B, whether object A will collide with object B).

4. It's not clear what will happen when the query points are out of images. What features shall we get?

5. It's not clear about the runtime of the LGP framework. Is it efficient to optimize or time-consuming?

6. The definition of "learned features" is not clear. Grasp feature and hang feature are pose distances between the current pose and the sampled success pose? Distance in SE3 or joint space? What about the collision feature?

7. The definition of the "feasible rate" (in table 1) is not clear.

8. For the "zero-shot imitation" experiment, where are the grid points placed? "centers of the target and the actual bounding balls" is not clear. Only the ball center is used?

9. For POSEICP, what's the result of only using Euclidean distance without feature space.



Minor comments:

1. Please adjust the presentation near equation (1) to make it look neat.

2. Page 5: "Similarly to the aforementioned image data augmentation", but data augmentation is not mentioned yet.

3. The location of the cameras is not clear. Where are they placed?

4. In table 1, why is PIFO (2 views) worse than PIFO?


[1] Yuan, Wentao, et al. "SORNet: Spatial Object-Centric Representations for Sequential Manipulation."

**Summary Of The Paper:**

The method proposes an implicit-field-function-based representation, which can be directly inferred from the camera images and be used for robot manipulation. The proposed method infers the object representation by querying about the implicit features of some pre-defined key points. The architecture projects the 3D query point into the 2D images and then exploits the pixel-wise features.

**Summary Of The Review:**

The authors propose a new method for capturing object representation directly from images and used for object manipulation. The idea of each component is not very novel. More importantly, the experiments miss baseline algorithms. I am not very convinced of the practical contribution or benefit of the proposed method. Can it solve more problems or outperform the existing method?

---

> ### Author Response · Authors · 2021-11-22
> **Thanks a lot for the detailed feedback! The additional experiments added**
>
> Thanks a lot for the detailed feedback!
>
> > [W1] The authors only compared their method with some ablated versions (i.e., global features, SDF representation) and didn't compare with other methods ... There are lots of strong baselines for the pick and place style tasks. What's the advantage over them, and why not compare with them?
>
> The primary motivation of this work is to eliminate the hand-engineering process in manipulation constraint modeling. We believe our experimental results show that such interaction constraints can be learned only from data (posed object images and feasible interaction poses). **-- EDIT -- We've also added the hand-designed constraints based on the mesh representation as a baseline. Please see the general comment above [[Link]](https://openreview.net/forum?id=I-nQMZfQz7F&noteId=Y9ni4ECqO2j).** Especially, the comparison to SDF representations highlights the benefits of task-guided representation learning over purely geometric representations. We also added a comparison of the proposed feature-based pose estimation method vs. the conventional ICP algorithm on point clouds where we found that, as already well-known, ICP easily gets stuck at local optima especially for the orientation estimation and the closest point matching based on the feature distance greatly alleviates the issue.
>
> Compared to other learning-based frameworks, the unique feature of this work is that we train interactions as constraint models which can be integrated into general sequential manipulation planning frameworks. Most existing works generate each interaction pose individually and later combine them, which is problematic for long-horizon planning, since combining individual samples from generative models implies a combinatorial complexity. For example, how to grasp a mug should be determined by how to hang it on the hook later, and vice versa. For the handover scenario, grasping, handover, and hanging poses as well as an entire trajectory to achieve those poses should be optimized altogether. We conducted additional experiments on inverse kinematics with generative models and found that only 14 out of 100 sets of individually generated poses were kinematically feasible for the handover scenario. The result is added in Section 6.2.
>
> Thank you for pointing us to the SORNet paper, which we added to the references. While it is definitively very interesting and relevant, our implicit object representations are used to model continuous interaction features and thereby to solve trajectory optimization, not to predict the symbolic predicates of the scene for task-level symbolic planning.
>
>
> > [W2] It's not clear how the learned features are incorporated into the LGP framework. What are the detailed examples of h_path, g_path, h_switch, and g_switch (equation (4)) for the grasp and hang experiments?
>
> We reformatted the paper so that Section 5 now can be devoted to the LGP integration of the learned feature. We believe the newly added Figure 3 helps the readers understand better as well.
>
> We devote Appendix A.2 to give concrete explanations on which manipulation constraints are imposed by the discrete actions and symbolic states. In short, the grasping and hanging actions impose three constraints each, about the learned interaction feature, zero-impact switching, and approaching direction. While a symbolic state indicates a mug is grasped by a gripper or hung on a hook, the static joint constraint is imposed on the mug frame. The collision constraint explained in A.3 is also included as a path constraint.
>
> > [W3] The current representation only cares about the relationship between the object and the pre-defined key points (e.g., gripper and fixed hook). It doesn't capture the relationship between the objects (e.g., whether object A is on top of object B, whether object A will collide with object B).
>
> This is a very good point. Yes, the current demos only handle the interactions between the robot vs. an object. The relationship between two or more objects can be considered by querying the representations of those objects at some pre-defined interaction region, either somewhere between the objects or (more naively) the entire workspace. We decided to focus on the interactions between the robot and an object and leave the others as future work because it already covers a wide range of manipulation tasks.
>
> Also, we would like to mention that the proposed manipulation framework allows for directly incorporating many object-object interactions, like touching, inserting, placing, and pushing, based on the learned SDFs by introducing the notion of “point of attack” as [1, 2] have proposed.
>
> [1] Toussaint, Marc, et al. "Describing physics for physical reasoning: Force-based sequential manipulation planning." RA-L 2020.
>
> [2] Driess, Danny, et al. "Learning Models as Functionals of Signed-Distance Fields for Manipulation Planning."  CORL 2021.

---

> > ### Author Response · Authors · 2021-11-22
> > **continue**
> >
> > > [W4] It's not clear what will happen when the query points are out of images. What features shall we get?
> >
> > As shown in Figure 2, the local image feature is concatenated with the coordinate feature which enables the out-of-image prediction. Although the coordinate feature alone would provide a little information, assuming the interactions happen around an object, these predictions outside of image regions need not be very precise. Moreover, the data sampling in Section 4.2 is encouraged to stay closer to the object.
> >
> >
> > > [W5] It's not clear about the runtime of the LGP framework. Is it efficient to optimize or time-consuming?
> >
> > LGP optimizes the trajectory by exploiting the sparse structure of the Hessian and the efficient Gauss-Newton method. For example, the three-mug and handover scenarios have a 1071-dimensional (one 7DOF arm for 60 steps and one 7DOF mug for 51, 31, 11 steps = 1071, the mug's rigid transformations before grasped are not included in optimization) and a 567-dimensional decision variable (two 7DOF arms for 30 steps and one 7DOF mug for 21 steps = 567), respectively, and are solved within a couple of minutes on a standard laptop even without highly optimizing the code (in terms of the GPU parallelization).
> >
> > We also would like to emphasize that, for planning, the images need to be encoded only once before the optimization begins. Thus the feature prediction during optimization only requires bi-linear interpolations for the local feature extraction and forward paths of a few layers of MLP.
> >
> >
> > > [W6] The definition of "learned features" is not clear. Grasp feature and hang feature are pose distances between the current pose and the sampled success pose? Distance in SE3 or joint space?
> >
> > “learned features” mean the entire interaction feature prediction scheme in (new) Figure 1 - we explicitly defined it in the revised introduction. The distance is measured in SE(3) and we added the statements in the paper.
> >
> > > [W6-2] What about the collision feature?
> >
> > The collision feature is built upon the learned SDF feature. We discuss in Appendix A.3 how we compute the pair distance between PIFO and obj$\in${sphere, capsule, convex mesh, PIFO}.
> >
> > > [W7] The definition of the "feasible rate" (in table 1) is not clear.
> >
> > By “feasible rate”, we meant the success rate of the optimized poses when simulated in Bullet. We renamed it to the success rate in the paper.
> >
> > > [W8] For the "zero-shot imitation" experiment, where are the grid points placed? "centers of the target and the actual bounding balls" is not clear. Only the ball center is used?
> >
> > We added Figure 20 which illustrates the grid points. The centers of the grid points are placed at the centers of the bounding balls.
> >
> >
> > > [W9] For POSEICP, what's the result of only using Euclidean distance without feature space.
> >
> > Thanks for the suggestion. We added the comparison with the conventional ICP algorithm on the 6D pose estimation problems. We found that, as already well-known, ICP easily gets stuck at local optima especially for its orientation estimation and the closest point matching based on the feature distance greatly alleviates the issue. The result can be found in Section 6.3.
> >
> >
> > > [C1, 2] Please adjust the presentation near equation (1) to make it look neat. Page 5: "Similarly to the aforementioned image data augmentation", but data augmentation is not mentioned yet.
> >
> > Thanks. We addressed the issues in the revision.
> >
> > > [C3] The location of the cameras is not clear. Where are they placed?
> >
> > We modified Figure 12(a) to show the cameras more clearly. Their poses are depicted as coordinate systems where the origin is the camera location, $-z$ axis (blue) is pointing the view direction, and $x$ and $-y$ axes (red and blue) are the directions of (u,v) coordinate of images.
> >
> > > [C4] In table 1, why is PIFO (2 views) worse than PIFO?
> >
> > The default setting has 4 views and so the performance improves with more views. We modified the manuscript to get rid of confusion.

---

> > > ### Author Response · Authors · 2021-11-22
> > > **continue**
> > >
> > > > **Summary Of The Review:**
> > > The authors propose a new method for capturing object representation directly from images and used for object manipulation. The idea of each component is not very novel. More importantly, the experiments miss baseline algorithms. I am not very convinced of the practical contribution or benefit of the proposed method. Can it solve more problems or outperform the existing method?
> > >
> > > While the object representation architecture can be seen as a simple adoption of the existing NIF architecture, we want to argue that its usage in the overall feature prediction, which computes the interaction feature which is a function of a 6D pose through the representations in the 3D space using key interaction points, and its training method, where the representation is shared across the tasks and jointly optimized with the head networks, are novel. **--- EDIT --- The additional comparison to the hand-designed features also highlights the advantage of our data-driven appoach.**
> > >
> > > In addition, integrating the learned features into LGP is new and gives enormous flexibility in generating diverse manipulation plans. The practical implication of it is to enable sensor-based planning with learned interaction features, without having object models. The additional experiments on IK with generative models especially highlight the advantages of the proposed planning framework in regard to the combinatoric complexity of the long horizon problems.
> > >
> > > Thanks again for your constructive reviews.

---

> > > > ### Comment · Reviewer_seqp · 2021-11-30
> > > > **keep original score**
> > > >
> > > > Thank authors for the detailed response and additional experiments, which have partially addressed my concerns. However, the experiments are critical to the main argument, but they didn't appear in the original submission. Also, the presentation of the paper has changed a lot. I prefer to see the revised paper in the later conferences with another thorough review. Taking other reviewers' scores into account (no strong accept), I would like to keep my original score.

---

### Official Review · Reviewer_gsUt · 2021-10-31

**Correctness:** 2
**Technical Novelty And Significance:** 1
**Empirical Novelty And Significance:** 2
**Recommendation:** 1
**Confidence:** 4

**Main Review:**

Strengths:
1. The motivation from neural implicit functions is interesting.
2. The videos show that the method works in the mug hanging task.

Weaknesses:
1. This paper is very difficult to follow.
2. It is very difficult to understand the framework by looking at Figure 1. Specifically, there seems to have 3 blocks in Figure 1, but in Section 3.1, the authors stated that the proposed framework consists of two parts. This is confusing and makes readers hard to follow.
3. In the introduction, what does "key interaction points which are rigidly attached to the robot frame" means? How can a point attach to a frame?
4. The related work section is too long (more than 1.5 pages).
5. The method is only trained on different mugs. It is unclear how well the method would perform if given data of other categories.
6. Since this paper studies a robotics problem, it is not convincing without showing video results using real robots.
7. In Figure 1, I do not understand how to compute forward kinematics for a given robot pose. Forward kinematics are typically computed when given joint angles for deriving poses. I believe this is a clear technical error.
8. The authors mentioned in Section 4.1 that the posed image data consists of 100 images. The authors should include an ablation study that analyzes how many images are needed. Is 100 sufficient?
9. How are ground-truth SDF values computed? The authors did not mention nor providing references in Section 4.1.
10. Since the method is only trained on 131 mesh models (as mentioned in Section 4.1) which is a very small dataset, I wonder if there is a possibility that overfitting happens.

**Summary Of The Paper:**

This paper studies the problem of learning representations for robotic manipulation tasks. The authors developed a method that represents objects as neural implicit functions. To method is trained in a data-driven fashion that learns the training pipeline from camera images to interaction features end-to-end. At each time step in a motion planning task, the implicit function is queried at interaction points that are attached to the robot frame. With the pixel-aligned representations, the representations at a certain spatial location correspond to the associated pixels of the images. Experiments are conducted on a mug hanging task. Videos in simulation environments show the effectiveness of the proposed method.

**Summary Of The Review:**

Although the problem studies by this submission is interesting, this paper is very difficult to follow. There are some details missing and some technical errors in the paper. There are also some unclear details and missing experiments mentioned in the above section.

---

> ### Author Response · Authors · 2021-11-22
> **Thank you for your feedback. We've revised the paper to make it easier to follow.**
>
> Thank you for your feedback!
>
> > [W1, 2] This paper is very difficult to follow. It is very difficult to understand the framework by looking at Figure 1. Specifically, there seems to have 3 blocks in Figure 1, but in Section 3.1, the authors stated that the proposed framework consists of two parts. This is confusing and makes readers hard to follow.
>
> We agree that the structure of the initial submission can be confusing to readers, so we reformatted the paper as follows: Section 3 is now solely about the task feature prediction which is depicted in Figure 1, and Section 5 discusses the actual manipulation planning where we added the overview of how the learned feature is integrated into LGP as well as Figure 3 which shows how the various components come together.
>
> Especially, we modified Figure 1 to have two blocks to make it aligned with the statements in Section 3. The rest of the whole framework is illustrated in Figure 3.
>
> > [W3] In the introduction, what does "key interaction points which are rigidly attached to the robot frame" means? How can a point attach to a frame?
>
> Figure 11 shows sets of grasping and hanging points consumed by PIFO. These points are fictitiously attached to the gripper or the hook frames and, by “rigidly attached”, we mean these points are moving together with the gripper or hook frames. As shown in Figure 3, during manipulation planning, the forward kinematics compute the pose of the frames from which the corresponding positions of key points can be obtained.
>
>
> > [W4] The related work section is too long (more than 1.5 pages).
>
> We have shortened the related work section.
>
> > [W5] The method is only trained on different mugs. It is unclear how well the method would perform if given data of other categories.
>
> In this work, we viewed a category as a group of objects that can be handled in the same way, i.e., the same set of tasks can be performed. While we can relax this condition and train the network on more general tasks, we restrict ourselves to category-level manipulation because the main contribution is learning implicit representations from the tasks. For example, it would have been possible to train a grasping feature network for more general objects if we had considered an object category of “graspable objects”.
>
> > [W6] Since this paper studies a robotics problem, it is not convincing without showing video results using real robots.
>
> We believe that sequential manipulation planning is a fundamental challenge in robotics and current approaches are limited by the challenges of representation, perception, and hand-engineering features. While other papers focus and demonstrate the execution of such plans on real robots, our paper aims to tackle these fundamental challenges to enable sensor-based planning with learned interaction features.
>
> > [W7] In Figure 1, I do not understand how to compute forward kinematics for a given robot pose. Forward kinematics are typically computed when given joint angles for deriving poses. I believe this is a clear technical error.
>
> By “robot pose”, we actually meant the robot joint configuration and the forward kinematics module in the original figure computes the keypoints’ positions for a given joint configuration. We renamed it “robot configuration” in the paper (since there can be prismatic joints as well).
> We believe the modified Figures 1 and 3 illustrate this more clearly.
>
> > [W8] The authors mentioned in Section 4.1 that the posed image data consists of 100 images. The authors should include an ablation study that analyzes how many images are needed. Is 100 sufficient?
>
> Our initial experiments were run with 200 images per object but we empirically found that 100 images resulted in the same level of performance. You can see from Figure 8(d-e) that 100 images provide a sufficient variety of objects’ appearances from different angles. The 3D priors used in our framework should enable PIFO to “interpolate” those appearances as well.
>
> In fact, just because we created 100 images, doesn’t mean the network will see only those images. The camera’s relative pose w.r.t. its viewing center has 6 DOFs and our proposed data augmentation ensures the augmented images span all those DOFs.
>
> > [W9] How are ground-truth SDF values computed? The authors did not mention nor providing references in Section 4.1.
>
> The ShapeNet dataset is given as triangular meshes and there are many open source libraries to compute SDFs. To robustify the procedure (as some meshes are not water-tight), we used the mesh_to_sdf library. We added the reference in the paper.

---

> > ### Author Response · Authors · 2021-11-22
> > **continue**
> >
> > > [W10] Since the method is only trained on 131 mesh models (as mentioned in Section 4.1) which is a very small dataset, I wonder if there is a possibility that overfitting happens.
> >
> > As we stated in Section 4.1, the 131 models were split into 78 training, 25 validation, and 28 test sets and we showed that, although the training data is small, our method generalizes, hence no overfitting happens. Further, as shown in Figure 4, it also works for even more out of distribution shapes, further showing that overfitting does not seem to be an issue.
> >
> > > **Summary Of The Review:**
> > Although the problem studies by this submission is interesting, this paper is very difficult to follow. There are some details missing and some technical errors in the paper. There are also some unclear details and missing experiments mentioned in the above section.
> >
> > We believe that the revision has made the paper much easier to follow. As summarized in the introduction, the main contributions of this work are (i) to represent objects as neural implicit functions upon which interaction features are trained, (ii) an image-based manipulation planning framework with the learned features as constraints, (iii) comparison to non-pixel-aligned, non-implicit function, and geometric representations, (iv) demonstration in various manipulation scenarios ranging from simple pick-and-hang to longer-horizon manipulations and zero-shot imitations.
> >
> > The missing details are filled in and the additional experiments on IK with generative models and pose estimation of conventional ICP back up the advantages of the proposed framework. We want to stress that misunderstanding about the forward kinematics was due to the confusing terminology and not “technical errors”, and the revised manuscript has resolved those confusions.

---

> ### Comment · Area_Chair_rYBJ · 2021-11-30
> **Responding to the revision**
>
> Reviewer gsUt,
>
> Thanks for your review. The authors have provided a rebuttal. Would you like to review it and see if it has addressed your comments?
>
> Thanks,
> AC

---

### Official Review · Reviewer_zYJc · 2021-11-01

**Correctness:** 4
**Technical Novelty And Significance:** 3
**Empirical Novelty And Significance:** 2
**Recommendation:** 6
**Confidence:** 3

**Main Review:**

Strengths
+ The idea is interesting and a natural progression from recent progress in 3D vision/representation learning and is a welcome change of pace where a vision problem is contextualized in a robotic task and developed with the intent to be a part of the full system.
+ The related work is well covered and the motivation is setup clearly i.e. we need vision backbones that work for manipulation tasks.

Weakness/Comments
- The interesting task-specific and task-agnostic dynamic is brought up but not really explored within the paper in the context of representation learning. Seems like a missed opportunity.
- While the broad strokes are clear (Fig 1, 2, Eq 4, etc) it was a bit hard to parse how the various components came together, in particular when instantiated given a particular task. At inference time, does the planning problem just take one image as input? is the image ego-centric? what are other inputs and do any of them fall under assumptions (like a 3D environment representation).
- Some more algorithmic details would be helpful. How to choose what set of points are used to condition PIFO and what if this choice changes between tasks? Could something like Mask-RCNN work for object segmentation vs Eq 5?
- Some more experiment/result details would be helpful. Why 2 view task performance drops compared to (1 view?) PIFO? In the Zero-shot imitation what's 'unseen' in test setting, new object geometry, anything else?

**Summary Of The Paper:**

This paper build on recent progress in implicit object representations and where the representations are trained(or fine-tuned) along with a feature head on down stream manipulation tasks. Experiments are conducted by using this trained representation within an LGP formulation solved with Gauss-Newton optimizer to find plans for manipulation problems.

**Summary Of The Review:**

Interesting idea and application, results are promising, but the write-up could be improved to understand the approach clearly.

---

> ### Author Response · Authors · 2021-11-22
> **Thanks a lot. The paper has been revised for a better exposition of the ideas.**
>
> Thanks a lot for your feekback!
>
> > [W1] The interesting task-specific and task-agnostic dynamic is brought up but not really explored within the paper in the context of representation learning. Seems like a missed opportunity.
>
> Thank you for raising an interesting point. First, the SDF representation baseline in Section 6.1 highlights the benefit of the task-specific representation: it is shown that, for the same task-head network architecture, the representation learned by PIFO results in much better task performance than the representation only trained by geometry data.
>
> The PCA visualizations in Figures 14 and 15 show the task-agnostic aspect of the learned representation: even though it was trained only by the tasks, the representation encodes the semantics of the objects which are consistent across the different objects. This task-agnostic aspect enables the learned representation to be used for (newly added) 6D pose estimation as well as zero-shot imitation.
> We revised the manuscript to show the above more clearly.
>
>
> > [W2] While the broad strokes are clear (Fig 1, 2, Eq 4, etc) it was a bit hard to parse how the various components came together, in particular when instantiated given a particular task.
>
> So as to describe better how the whole system works, we reformatted the paper as follows: Section 3 is now solely about the task feature prediction (as shown in Figure 1), and Section 5 discusses the actual manipulation planning where we added the overview of how the learned feature is integrated into LGP as well as Figure 3 which shows how the various components come together.
> In addition, we added concrete explanations on the manipulation constraints and the collision features in Appendix A.2 and A.3, respectively.
>
> > [W2-2] At inference time, does the planning problem just take one image as input? is the image ego-centric? what are other inputs and do any of them fall under assumptions (like a 3D environment representation)?
>
> All the considered manipulation scenes have four cameras and the raw images contain the entire scene (the camera poses can be seen in Figure 12(a)). The multi-view processing in Section 5.1 transforms these raw images into object-centric ones and computes the corresponding camera extrinsics/intrinsics. Figure 12 shows how this process works. In particular, no further object representations like its 3D model/mesh are necessary for trajectory optimization.
>
> > [W3] Some more algorithmic details would be helpful. How to choose what set of points are used to condition PIFO and what if this choice changes between tasks?
>
> We believe the modified manuscript provides a clearer overview. Figure 11 shows sets of grasping and hanging points consumed by PIFO; these points are fixed within a task and vary across different tasks. In theory, three points that are not collinear should be sufficient to represent the frame’s pose, but the feature head can learn how to consume representation vectors even when they were fed from more (redundant) key points. The keypoints can also be optimized with the network, but we empirically couldn’t find any improvements from that.
>
> > [W3-2] Could something like Mask-RCNN work for object segmentation vs Eq 5?
>
> The multi-view processing in Section 5.1 requires the object masks which Mask-RCNN can provide. This work assumes the object masks are available with the raw images and camera parameters.
>
> > [W4] Some more experiment/result details would be helpful. Why 2 view task performance drops compared to (1 view?) PIFO? In the Zero-shot imitation what's 'unseen' in test setting, new object geometry, anything else?
>
> We also revised the experiment section to expose the setup/result more clearly.
> The default PIFO gets 4 views and so the performance improves with more views.
> The new object geometry is the only unseen aspect in the zero-shot imitation. The point we try to make here is that the learned representation from images allows for transferring 6D poses of one object to other objects, without defining any canonical object coordinate systems.
>
> This is closely related to category-level 6D pose estimation problems. Thus we added the comparison of our method (FCP) with the conventional ICP based on point clouds. It is well known that ICP easily gets stuck at local optima especially for the orientation estimation and our experiment shows that the closest point matching based on the learned feature greatly alleviates the issue.
>
> > **Summary Of The Review:**
> Interesting idea and application, results are promising, but the write-up could be improved to understand the approach clearly.
>
> We believe that the revised manuscript exposes the main contributions of this work more clearly and the additional experiments on IK with generative models and pose estimation of conventional ICP highlight the advantages of the proposed framework over the existing works. Thank you again for your constructive comments.

---

> > ### Comment · Reviewer_zYJc · 2021-11-29
> > **Thank you for the response**
> >
> > The paper reads much better and is more easier to follow the overall approach. Thanks for the additional explanation and the new experiments based on other reviews. After these revisions, there is sufficient contributions here of value to the community, thus I am increasing my score from 5 to 6 (I would select 7 but there isn't an option for it).

---

> > > ### Author Response · Authors · 2021-11-30
> > > **Thank you for raising the score!**
> > >
> > > Thank you for raising the score! Your comments were very helpful when modifying the flow of the manuscript.
> > >
> > > If 7 isn't in the options, would there be any chance to go for 8 instead of 6? It could make a big difference since the scores are on the borderline and some reviewers haven't responded yet. (Also we'd like to ask if your current scores for "Technical Novelty" are up-to-date.) Compared to the current version, we're planning to include the comparison with the hand-engineered features on meshes, polish the paper further for even better exposition and create a supplementary video as a better visual aid for the camera-ready submission.

---

### Author Response · Authors · 2021-11-22
**Overview of the revision**

We thank the reviewers for their helpful comments and suggestions!

Before replying in detail to the reviewers’ comments, we would like to highlight a point that seemingly was unclear but is at the core of our work. The task features we use are trained based on empirical data of successful interactions in random trials, thereby replacing the human expertise to hand-design interaction features. These features are trained as heads on top of and jointly with the implicit functional object representation. Therefore this work enables sensor-based planning which doesn't rely on the traditional perception of geometric representation, nor hand-engineered manipulation constraints. To summarize again, the novel technical contributions are (i) representing objects as neural implicit functions upon which the manipulation constraints are learned, (ii) jointly training the representation and constraint models, and (iii) integrating the learned models in the sensor-based manipulation planning framework.

In the following, we summarize the major revision points we’ve made. We believe that the revised manuscript exposes our main contributions more clearly and the additional experiments highlight the advantages of the proposed framework over existing works.

1. **Reformatting the paper:**
To describe better how the whole system works, we reformatted the paper as follows: Section 3 is now solely about the task feature prediction and Section 5 discusses the actual manipulation planning where we added the overview of how the learned features are integrated into LGP. Figures 1 and 3 illustrate what each section addresses more clearly.
In addition to the high-level description in Section 5.1, Appendix A.2 gives more concrete explanations on which manipulation constraints are imposed by the discrete actions and symbolic states. In short, the grasping and hanging actions impose three constraints each, about the learned interaction feature, zero-impact switching, and approaching direction. While a symbolic state indicates a mug is grasped by a gripper or hung on a hook, the static joint constraint is imposed on the mug frame.
In Appendix A.3, we discuss how the collision constraint is built upon the learned SDF feature and enumerate the concrete cases of handling pair distances of PIFO vs. obj$\in${sphere, capsule, convex mesh, PIFO}.


2. **Experiments on utilizing generative models for inverse kinematics:**
Compared to other learning-based frameworks, the unique feature of this work is that we train interactions as constraint models which can be integrated into general sequential manipulation planning frameworks. Most existing works generate each interaction pose individually and later combine them, which is problematic for long-horizon sequential manipulation planning, since combining individual samples from generative models implies a combinatorial complexity. For example, how to grasp a mug should be determined by how to hang it on the hook later, and vice versa. For the handover scenario, grasping, handover, and hanging poses as well as an entire trajectory to achieve those poses should be optimized altogether. We conducted additional experiments on inverse kinematics with generative models and found that only 14 out of 100 sets of individually generated poses were kinematically feasible for the handover scenario. The result is added in Section 6.2 and Figures 18-19.


3. **Comparison to the conventional ICP algorithm on 6D pose estimation:**
The revision is also including a comparison of the proposed feature-based 6D pose estimation method vs. the conventional ICP algorithm on point clouds where we found that, as already well-known, ICP easily gets stuck at local optima, especially for its orientation estimation. In contrast, the closest point matching based on the feature distance using our approach greatly alleviates the issue. The result can be found in Section 6.3 and Figures 20-21.

---

> ### Author Response · Authors · 2021-11-29
> **Additional expeirments added!**
>
> Thanks to the additional feedback from *Reviewer 23pA* as well as a review from *Reviewer seqp*, we conducted additional experiments on the traditional mesh representations + hand-engineered interaction constraints (The details can be found below). These features are evaluated with the meshes reconstructed by the learned SDF representations (4 views) as well as with the ground-truth meshes. The grasping success rates (Training / Test mugs) were **62.8% / 75.0%** on the GT meshes and **66.7% / 42.9%** on the reconstructed meshes. The hanging success rates were **94.9% / 92.9%** on the GT meshes and **78.2% / 60.7%** on the reconstructed meshes. The comparisons with other representations can be easily seen in this table [[Link]](https://drive.google.com/file/d/1LYFjxG-BU3wZQWbAjlCKUWYG-jQg4KW1/view?usp=sharing).
>
> As shown in the figures of some failure cases [[Link]](https://drive.google.com/file/d/16cmhJsLiyqnUNBEonEqDnJXP2z__gkGL/view?usp=sharing), the reconstruction error of a mesh is directly associated with the planning result; e.g. it would never grasp not-reconstructed parts, would try to hang the mug through a wrongly-generated hole, and/or could result in collisions. While the perception pipeline for this representation is never encouraged to reconstruct the “graspable/hangable parts” more accurately, we can view our end-to-end representation learning via task supervision as a way to do so. Moreover, the hand-engineered feature sometimes produces a wrong grasping pose even for the ground truth mesh (e.g., (a) in the figure above). One can argue that a better interaction feature could be hand-designed by investigating the physics and kinematic structures more deeply, but that would require a huge amount of human insights/efforts and thus is less scalable. In contrast, our data-driven approach eliminates this procedure and directly learns the interaction constraint models from empirical success data of physical interactions.
>
> We think these new results have justified the necessity and advantage of our data-driven approach and, with other experimental results, the hypotheses made in the paper are now well supported. We also strongly believe that this work is well-aligned with the theme of ICLR, highlighting yet another motivation of learning representations, and is an important improvement in robotics as well.
>
> **Experimental details:**
> Throughout the experiments, objects are represented by meshes, especially with the convex-decomposition (https://github.com/kmammou/v-hacd) for non-convex shapes, and thus pair-distance and collision between meshes can be computed via the GJK/MPR algorithm. On top of this mesh representation, the grasping and hanging constraints are defined and optimized as follows:
> - The grasping constraint consists of the aforementioned collision constraints and the so-called oppose constraint. The oppose feature takes as input three meshes, finger1, finger2, and (a set of decomposed) object meshes to grasp. It computes the minimum pair-distances from finger1 and finger2 to the object and returns the sum of two vectors, i.e., (finger1->object + finger2->object). Making the oppose feature (0,0,0) places the object in the middle of two fingers with proper orientation. This is a widely used hand-engineered constraint that works very well for simple shapes, such as spheres, capsules, etc. Because the mug shapes are highly non-convex we ran the optimization from 100 initial seeds and took the best one with the minimum constraint violation.
> - Given the object mesh, the hanging feature iteratively generates a collision-free pose (up to 10,000 iterations) and checks if the hook is kinematically trapped by the mug (as done in data generation). If trapped, it returns the pose difference so that optimizer can output the found pose.

---

### Decision · Program_Chairs · 2022-01-20

**Decision:**

Reject

**Comment:**

The paper initially received negative reviews; the authors did a good job during the response period: two reviewers have updated their scores to 6. The AC has carefully read the reviews, responses, and discussions, and agreed that the authors have also mostly addressed the concerns of reviewer gsUt as well. It is unprofessional for reviewer gsUt to not engage in discussions after multiple requests.

The AC however also agrees with reviewer seqp that the new changes are major, and submissions are supposed to be evaluated in their initial form. Further, neither of the positive reviewers would like to champion the paper.

The final recommendation is to reject the paper. The authors are encouraged to further improve and flesh out the paper based on the reviews for the next venue.